

# 1  Orbital CO₂ reconstruction using boron isotopes during the
# 2  late Pleistocene, an assessment of accuracy.

Elwyn de la Vega[a,b], Thomas B. Chalk[a,c], Mathis P. Hain[d], Megan R. Wilding[a], Daniel Casey[a], Robin
Gledhill[a], Chongguang Luo[a,e], Paul A. Wilson[a], Gavin L. Foster[a].
[a]School of Ocean and Earth Science, National Oceanography Centre Southampton, University of
Southampton, Waterfront campus, Southampton SO14 3ZH, UK.
[b]University of Galway, Ollscoil na Gaillimhe, department of Geography, University Road, Galway,
H91 TK33, Ireland.
[c]Centre Européen de Recherche et d'Enseignement des Géosciences de l'Environnement (CEREGE),
Bâtiment Pasteur, Europole Mediterraneen de l'Arbois BP80,13545 Aix-en-Provence cedex 4.
[d]Earth and Planetary Sciences, University of California, Santa Cruz, CA, USA.
[e]State Key Laboratory of Ore Deposit Geochemistry, Institute of Geochemistry, Chinese Academy of
Sciences, Guiyang 550081, P.R. China.
Correspondence to Elwyn de la Vega: elwyn.delavega@universityofgalway.ie
**Abstract.**
Boron isotopes in planktonic foraminifera are a widely used proxy to determine ancient surface seawater
pH, and by extension atmospheric $CO_2$ concentration and climate forcing on geological time scales.
Yet, to reconstruct absolute values for pH and $CO_2$, we require a $\delta^{11}B_{foram-borate}$ to pH calibration and
independent determinations of ocean temperature, salinity, a second carbonate parameter, and the boron
isotope composition of seawater. Although $\delta^{11}B$-derived records of atmospheric $CO_2$ have been shown
to perform well against ice core-based $CO_2$ reconstructions, these tests have been performed at only a
few locations and with limited temporal resolution. Here we present two highly resolved $CO_2$ records
for the late Pleistocene from ODP Sites 999 and 871. Our $\delta^{11}B$-derived $CO_2$ record shows a very good
agreement with the ice core $CO_2$ record with an average offset of 4.6 ± 49 (2σ) ppm, and a RMSE of
25 ppm, with minor short-lived overestimations of $CO_2$ (of up to ~50 ppm) occurring during some
glacial onsets. We explore potential drivers of this disagreement and conclude that partial dissolution
of foraminifera has a minimal effect on the $CO_2$ offset. We also observe that the general agreement
between $\delta^{11}B$ -derived and ice core $CO_2$ is improved by optimising the $\delta^{11}B_{foram-borate}$ calibration. Despite
these minor issues a strong linear relationship between relative change in climate forcing from $CO_2$
(from ice core data) and pH change (from $\delta^{11}B$) exists over the late Pleistocene, confirming that pH
change is a robust proxy of climate forcing over relatively short (<1 million year) intervals. Overall,
these findings demonstrate that the boron isotope proxy is a reliable indicator of $CO_2$ beyond the reach
of the ice cores and can help improve determinations of climate sensitivity for ancient time intervals.

**1- Introduction.**
The boron isotope composition of ancient planktonic foraminifera shells is widely used to reconstruct
past concentrations of atmospheric $CO_2$ to understand the drivers and responses of climate change over
orbital and geological time scales. Unlike many environmental proxies where it is difficult to assess the
accuracy of the resulting reconstructions (e.g. for sea surface temperature), the boron isotope pH/$CO_2$
proxy can directly be compared with the ice core $CO_2$ records, i.e. the West Antarctic ice sheet divide
(Ahn et al., 2012), the EPICA (European Project for Ice Coring in Antarctica) dome Concordia ice core
record (Siegenthaler et al., 2005; Luthi et al., 2008; Bereiter et al., 2015), and the Vostock ice core
record (Petit et al., 1999). This comparison of $CO_2$ over the last 800 kyr provides a very powerful test
of proxy accuracy. Several past intervals have been studied to test the boron isotope proxy in this way





(Sanyal et al., 1995; Foster, 2008; Hönisch and Hemming, 2005; Henehan et al., 2013; Raitszch et al.,
46   2018).

Given the success of these comparisons, the boron isotope proxy has been used to investigate the
interaction between CO2, the ocean carbon cycle and climate beyond the reach of the ice cores, such as
during the Mid-Pleistocene transition (Hönisch et al., 2009; Chalk et al., 2017; Dyez et al., 2018), the
Pliocene (Martinez-Boti et al., 2015, de la Vega et al., 2020), the Miocene (Foster et al., 2012; Greenop
et al., 2014, Guillermic et al., 2022), the Eocene (Anagnostou et al., 2016, 2020; Harper et al., 2020),
Paleocene-Eocene boundary (Penman et al., 2014; Gutjahr et al. 2017) and the Cretaceous-Palaeogene
boundary (Henehan et al., 2019). Application of the boron isotope proxy is however complicated by the
need for: (i) an empirical species-specific calibration of $\delta^{11}B_{foraminifera}$ to $\delta^{11}B_{borate}$ in the pH expression
(Henehan et al., 2013, 2016, hereafter $\delta^{11}B_{foram-borate}$ calibration), sometimes including extinct species
for deep-time reconstruction; (ii) $\delta^{11}B$ of seawater ($\delta^{11}B_{sw}$), temperature and salinity in the past to
calculate pH from $\delta^{11}B$; and (iii) a second carbonate parameter (typically total alkalinity, total dissolved
inorganic carbon, DIC, or calcite saturation state) to convert pH to $CO_2$. While these variables do not
influence the magnitude of uncertainty equally in all time intervals, assessment of the boron-based
reconstructions against existing ice-core records is a powerful test of the proxy's accuracy.
Recently, Hain et al. (2018) suggested that the radiative forcing from $CO_2$ change ($\Delta F_{CO2}$) is linearly
related to pH change ($\Delta pH$) of equilibrated water of the low-latitude surface ocean when the $CO_2$ change
occurs faster than the residence time of carbon with respect to silicate weathering (e.g., ~1 million years
(Myr)). That is, glacial/interglacial $CO_2$ climate forcing could be estimated directly from reconstructed
$\Delta pH$. Given that one of the main priorities for accurate reconstructions of past $CO_2$ levels is to allow
determinations of climate sensitivity, defined as the temperature response to a radiative forcing –
typically a doubling of $CO_2$ with associated slow and fast feedbacks (e.g. Rohling et al., 2013, 2018) –
this recognition may provide a useful shortcut. Climate forcing is a perturbation of the planet's energy
balance averaged over the planet (Hansen et al., 2008) and $CO_2$ forcing, $\Delta F_{CO2}$ expressed in $W.m^{-2}$, at a
given time can be written as:
$$\Delta F_{CO2} \cong \alpha_{2xCO2} * \frac{\Delta \log_{10} CO_2}{\log_{10} 2} \ (1)$$
where $\alpha_{2xCO2}$ is the sensitivity of the radiative balance per doubling of $CO_2$, and $\Delta\log_{10}CO_2$ is the $CO_2$
change over time expressed in terms of how many 10-foldings of proportional (not absolute) $CO_2$
change  (Hain et al., 2018).
By considering basic equilibrium reactions of carbon species, $\Delta\log_{10}CO_2$ can be derived and expressed
as:

$$\Delta \log_{10} CO_2 \cong \Delta\log_{10} DIC + \Delta pK_0 + \Delta pK_1 - \Delta pH \ (2)$$
Hain et al. (2018) showed that the terms $\Delta\log_{10}DIC$ and $\Delta pK_0+\Delta pK_1$ are small and that $\Delta logCO_2$ can
therefore simply be expressed as :

$$\Delta \log_{10} CO_2 \cong -\Delta pH \ (3a)$$

$$\Delta F_{CO2} \cong -\frac{\log_{10} 2}{\alpha_{2xCO2}} \Delta pH \ \cong \ -12.3\Delta pH \ (3b)$$
To assess the uncertainty of this approximate -1:1 $\Delta\log_{10}CO_2/\Delta pH$ relationship Hain et al. (2018)
considered three different end-member causes to compute the accurate $\Delta\log_{10}CO_2/\Delta pH$ relationship: (1)
DIC addition/removal yields a slope of -1.3:1 (relative to the basic formalism), (2) $CaCO_3$
addition/removal (e.g. precipitation/dissolution, riverine input) yields a slope of -0.9:1, and (3)
warming/cooling yields a slope of -1.1:1. That is, even if $\Delta pH$ was known exactly this range of plausible
slopes results in estimated $\Delta\log_{10}CO_2$ and $\Delta F_{CO2}$ that are systematically biased by -10% for change
caused purely by $CaCO_3$ variations or +30% for change purely caused by DIC variations relative to the



approximate -1:1 $\Delta\log_{10}CO_2/\Delta pH$ relationship. While introducing such structural uncertainty in the
estimation of $\Delta F_{CO2}$ is a concern, this approach eliminates the need to assume a second carbonate system
parameter and the uncertainty incurred thereby. An estimate of $\delta^{11}B_{sw}$ is still needed to reconstruct pH
based on the boron isotope proxy system (Foster and Rae, 2016) but estimated pH change (i.e., $\Delta pH$) is
much less sensitive to error in assumed $\delta^{11}B_{sw}$ than is absolute pH (Hain et al., 2018). An important
caveat to estimating $\Delta F_{CO2}$ directly from $\Delta pH$ is that the intercept of the $\Delta\log_{10}CO_2/\Delta pH$ relationship
can change with silicate weathering carbon cycle dynamics thought to be important on a million year
timescale, such that the approach is applicable for orbital timescale variability and short-term shifts but
not for long-term trends in $\Delta F_{CO2}$. Therefore, the orbital timescale ice age cycles of atmospheric $CO_2$
reconstructed from air occluded in Antarctic ice cores offer a unique opportunity to determine the
$\Delta\log_{10}CO_2/\Delta pH$ relationship observationally and compare to theory. Furthermore, Hain et al., (2018)
raise the possibility that the $\Delta\log_{10}CO_2/\Delta pH$ relationship could be decomposed based on the different
end-member slopes to constrain the relative importance of the mechanism causing the pH and $CO_2$
changes.

In light of these recent advances, our aims here are twofold. First, we extend previous ice-core validation
studies (Foster, 2008; Henehan et al. 2013; Chalk et al., 2017) and test the extent to which boron
isotopes reconstruct $CO_2$ faithfully when current methods and assumptions are applied. In contrast to
most previous studies, we use two deep ocean sites and present $\delta^{11}B$ and $CO_2$ data at high temporal
resolution (1 sample every ~3 to 6 kyr). This enables: (i) a thorough test of the assumptions typically
made including the central tenet of atmospheric $CO_2$ proxies that surface ocean $CO_2$ remains in
equilibrium with the atmosphere over time at any given site, (ii) an evaluation of the overall uncertainty
of the proxy; (iii) an evaluation of the influence of variable foraminiferal preservation on the accuracy
of the $CO_2$ reconstructed; and (iv) a refinement of a number of the input assumptions and uncertainties,
including the $\delta^{11}B_{borate-foram}$ calibration. Second, we evaluate the approach of Hain et al. (2018) and
assess the robustness of pH change to not only provide insights into the magnitude of climate forcing
from $CO_2$ change, but also the ability of this approach to provide insights into the causes of $CO_2$ change
over glacial-interglacial cycles.
**2.     Methods.**
**2.1     Core location and oceanographic setting.**
To accurately reconstruct atmospheric $CO_2$ with the $\delta^{11}B$-$CO_2$ proxy, it is essential to measure $\delta^{11}B$ in
foraminifera from locations where the $CO_2$ flux between the ocean and the atmosphere is in near
equilibrium. We therefore target regions of the ocean where the water column is stratified and
oligotrophic as these regions are most likely to attain this condition (Takahashi et al., 2009). Here,
following previous studies (Foster, 2008, Henehan et al., 2013; Chalk et al., 2017), we report data from
ODP Site 999 (Figure 1, 12.75°N, 78.73°W, water depth 2827 m, sedimentation rate 3.7 cm/ky) in the
Caribbean and supplement this well studied site with samples from ODP Site 871 in the Western Pacific
(5.55°N, 172.35°E, water depth 1255m, sedimentation rate ~1 cm/ky). The sediments studied at ODP
Site 871 are shallowly buried and the site today features a deep thermocline and is located off the
equator, hence they are unlikely to be influenced by significant equatorial upwelling (Dyez and Ravelo,
2013, 2014). These two sites show a minor annual mean disequilibrium of +12 ppm (range ~0 to ~30
ppm, Takahashi et al., 2009) for ODP Site 871, and +21 ppm (Olsen et al., 2004; Foster, 2008) for ODP
Site 999. These disequilibria are used to correct our $CO_2$ data derived from $\delta^{11}B$ and are assumed to be
constant throughout the entire record presented here.




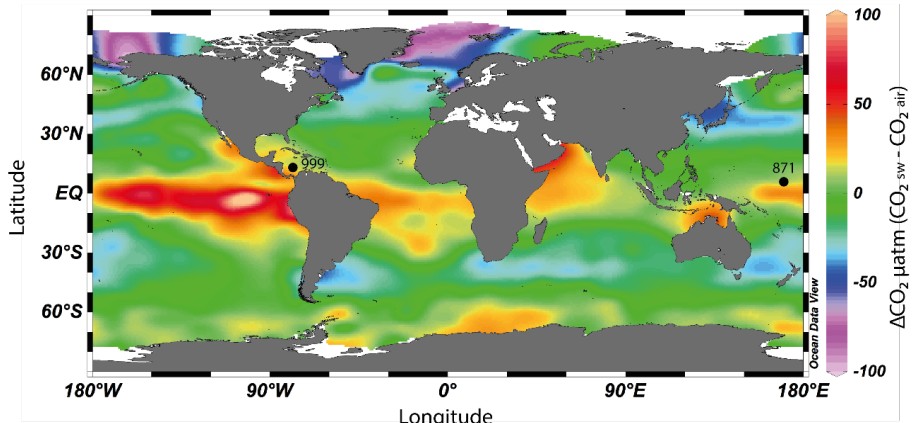

Figure 1. Map of air-sea $CO_2$ disequilibrium (seawater – air) in ppm and location of ODP sites used in this study.
$CO_2$ data from Takahashi et al. (2009). The map was made with Ocean Data View (Schlitzer, 2022).
**2.2 Samples.**
*2.2.1 Sample selection and preparation.*
Samples of deep-sea sediment from our two study sites were taken at 6cm (~3ky) and 10cm (~6ky)
resolution at ODP 871 and 999 respectively. Around 1-2 mg of the foraminifer (between 120 and 200
individuals) from the species *Globigerinoides ruber sensu stricto white* (here after *G. ruber ss*) were
hand-picked from the size fraction 300-355 4μm for a target of 10 to 20 ng of boron. *G. ruber ss* was
chosen here because it is readily identified, is abundant throughout our chosen time interval and a
$\delta^{11}B_{\text{foram-borate}}$ calibration that accounts for vital effects is available from culture, plankton tows and core-
top samples (Henehan et al., 2013). It is also known to live in the upper surface of the ocean with a
relatively small depth range which prevents significant influence of deeper more remineralised $CO_2$-
rich waters on the measured $\delta^{11}B$. The morphotype *G. ruber sensu lato* (hereafter *G. ruber sl*) has
slightly different morphology (Aurahs et al, 2001; Carter et al., 2017) and is thought to live in deeper
water compared to *G. ruber ss* (Wang, 2000). The morphotype *G. ruber sl* was also hand separated and
analysed at lower resolution at ODP 871 to monitor any change over time in morphotype differences in
$\delta^{11}B$ that could result from different habitats. For similar reasons, carbon and oxygen isotopes ($\delta^{18}O$
and $\delta^{13}C$) were also measured on *G. ruber ss* and *sl* for comparison on the whole record at ODP 871.
For this, around 10 individuals of *G. ruber* per sample were picked, their shells gently broken open and
mixed and then a 100 μg aliquot of the homogenised carbonate was measured using a Thermo KIEL IV
Carbonate device at the University of Southampton, Waterfront Campus.

*2.2.2 Age constraints.*
Samples were taken from 1.5 to 5 metres below sea floor (mbsf) for ODP 871 and from 9 to 21 mbsf
for ODP 999. Sample age at Site 871 was initially determined from sample depth using published age
models (Dyez and Ravelo, 2013). At Site 999, the age was determined by developing a new benthic
$\delta^{18}O$ record. The initial age model at Site 871 was refined by measuring $\delta^{18}O$ on the benthic species
*Cibicoides wuellerstorfi* (50 μg of 3-5 mixed, crushed and homogenised specimens) measured on a
Thermo KIEL IV Carbonate device at the University of Southampton, Waterfront Campus. These new
$\delta^{18}O$ data (Figure 2) were then tuned to the benthic $\delta^{18}O$ LR04 stack (Lisiecki and Raymo, 2005) using
Analyseries (Paillard et al., 1996).



*2.2.3    Fragment counts.*
Foraminifera fragment counts were conducted on ODP Site 871 to monitor variations in carbonate
preservation. Samples were sub-sampled using a splitter (in order to maintain homogeneity) and poured
onto a picking tray. The fragmentation index (FI) was calculated following the approach of Howard and
Prell (1994) and Berger (1970) where percentage fragment is defined as:

$$FI = 100 * \frac{\text{number of fragments}}{\text{number of fragments} + \text{number of whole tests}} \quad (4)$$

Counts of whole intact grains and fragments of grains were conducted three times and averaged. The
standard deviation ($1\sigma$) of the fragmentation index is 1.69. This approach followed that used in an early
study at ODP Site 999 (Schmidt et al. 2006) ensuring that the datasets between the two sites are
comparable
*2.2.4    Boron separation.*
The hand separated foraminifera tests for boron isotope analysis were broken open, detrital clay was
removed, and oxidatively cleaned and leached in a weak-acid to obtain a primary carbonate signal using
established methods (Barker et al., 2003). Samples were then slowly dissolved in ~ 100 µl 0.5M $HNO_3$
added to 200 µl of MQ water. Dissolved samples were then centrifuged for 5 minutes to separate any
remaining undissolved contaminants (e.g. silicate grains, pyrite crystals) and transferred to screw top 5
ml Teflon pots for subsequent boron separation. An aliquot equivalent to 7% of each sample was kept
for elemental analysis and transferred to acid cleaned plastic vials in 130 µl 0.5M $HNO_3$. Samples were
purified for boron using anion exchange column chemistry method prior to isotope analysis as described
elsewhere (Foster, 2008). A total procedure blank (TPB) was conducted for each batch of samples and
typically ranged from 0-50 pg which represents a very small contribution relative to our sample size (0-
0.25%), hence no samples required correction in this study.
**2.3   Effect of dissolution (leaching experiment).**
To investigate the effect of partial dissolution on measured $\delta^{11}B$, a leaching experiment was conducted
on two species of commonly analysed planktic foraminifera: *G. ruber ss* and *Trilobatus sacculifer*
(hereafter *T. sacculifer*). Around ~ 180 *G. ruber ss* (size 300-355 µm) and 40 *T. sacculifer* (size 500-
600 µm) were picked four times and the samples were treated like so: one split was the control and
received no treatment, and the three other samples (whole foraminifera) were placed in 0.0001 M Teflon
distilled $HNO_3$ (pH 4) for 2, 4, and 6 hours respectively. The experiment was repeated for *G. ruber ss*
by longer treatments, up to 10 hours in the dilute acid. The foraminifera subjected to these partial
dissolution tests were then treated using the same cleaning and chromatography protocols described
above.

We acknowledge that our leaching tests aren't as thorough as those described in some other studies
(e.g. Brown and Elderfield, 1996; Sadekov et al., 2010) but provide useful first-order insights into the
susceptibility of $\delta^{11}B$ to partial dissolution of foraminiferal tests.
**2.4   Analytical techniques**
Boron isotope analyses were performed on a ThermoScientific Neptune multi collector inductively
coupled plasma mass spectrometer (MC-ICPMS) with $10^{12}$ W amplifier resistors using a standard-
sample bracketing routine with NIST 951 boric acid standard (following Foster et al. 2013 and Foster,
2008). Elemental analysis was performed on each dissolved sample using a ThermoScientific Element
inductively coupled plasma mass spectrometer (ICPMS). All analyses were carried out at the University
of Southampton, Waterfront Campus (following Foster, 2008 and Henehan et al., 2015). Element to
calcium ratios were measured with $^{43}Ca$ and $^{48}Ca$ and measured against in house mixed element





standards. Elemental ratios measured included: B/Ca, Mg/Ca, Al/Ca, Mn/Ca, Sr/Ca. Based on the
reproducibility of our in-house standards, the uncertainty for most elemental ratios is ~ 5% (at 95%
confidence).

### 2.5 Constraints on δ¹¹B-derived pH and CO₂.

**2.5 Constraints on $\delta^{11}$B-derived pH and CO$_2$.**
*2.5.1    From $\delta^{11}B$ to pH.*
Seawater pH is related to the boron isotopic composition of dissolved borate ion by the following
equation:

$$pH = pK_B - \log\left(-\frac{\delta^{11}B_{sw} - \delta^{11}B_{borate}}{\delta^{11}B_{sw} - a_B * \delta^{11}B_{borate}(a_B - 1)}\right) \quad (5)$$


where the isotopic fractionation factor $\alpha_B$ between B(OH)$_3$ and B(OH)$_4^-$, is 1.0272 as determined by
Klochko et al. (2006) and the $\delta^{11}$B of seawater is 39.61 ‰ (Foster et al., 2010) for both sites and kept
constant throughout the record due to the long residence time of boron (10-20 Myrs, Lemarchand et al.
234 2002).


The sea surface temperature (SST) values necessary to calculate pK$_B$ in equation (5) were determined
at both sites using the Mg/Ca of *G. ruber* and the relationship of Anand et al. (2003):

$$SST = \frac{\ln\left(\frac{\frac{Mg}{Ca}_{surf}}{0.38(\pm 0.02)}\right)}{0.09(\pm 0.003)} \quad (6)$$


This calibration does not include a depth correction but yields temperatures from core top samples that
are consistent with modern SST (Olsen et al., 2016). The salinity that is used in the expression of pK$_B$
is kept constant for both sites (35 PSU) due to the very minor effect of salinity on calculated CO$_2$.

To investigate the effect of the recently proposed pH effect on reconstructed Mg/Ca-derived SST and
hence reconstructed CO$_2$, we've explored a scenario wherein we apply a pH correction on Mg/Ca-SST
using the iterative approach of Gray and Evans (2019).
*2.5.2    From pH to CO$_2$.*
Calculating CO$_2$ from boron isotope derived pH is dependent on the determination of a second
parameter of the carbonate system. Here we use the modern value of total alkalinity (TA) at each site:
2279 and 2350 μmol/kg at ODP 871 and ODP 999, respectively (Shipboard Scientific Party, 1993;
Takahashi et al., 2009). Following Chalk et al. (2017), these values were kept constant throughout the
whole record. To account for any variations in alkalinity, a generous uniform (or flat) uncertainty of
175 μmol/kg is applied (i.e. equal likelihood of values within the range of uncertainty). This range in
TA encompasses the likely range in this variable on glacial-interglacial (e.g. Toggweiler, 1999; Hain et
al., 2010; Cartapanis et al., 2018) or longer timescales (Hönisch et al. 2009), and its adoption means the
local site is not tied to a global sea-level record as had been practice previously. We avoid drawing this
link because the ~+3% (+68μmol/kg) concentration increase of solute alkalinity occurring from sea-
level lowering during the last glacial maximum may not have been the dominant driver of ocean
alkalinity change (Boyle, 1988a/b; Sigman et al., 1998; Toggweiler, 1999; Hain et al., 2010; Cartapanis
et al., 2018). By assuming a uniform distribution for TA we avoid imposing a temporal evolution to this
variable because evolution of TA through a glacial cycle is uncertain and is unlikely to be simply a
function of sea-level or salinity (e.g. Dyez et al. 2018) due to the effect of carbonate compensation.
The surface water CO$_2$ is then calculated as (Zeebe and Wolf-Gladrow, 2001):



$$CO_2 = \frac{TA - \frac{K_B * B_T}{K_B + [H^+]} - \frac{K_W}{[H^+]} + [H^+]}{\frac{K_1}{[H^+]} + \frac{2K_1 K_2}{[H^+]^2}} \quad (7)$$

where TA is the total alkalinity, $K_B$ the equilibrium constant of boron species in seawater, $B_T$ the
concentration of boron in seawater (432.6 μmol/kg, Lee et al., 2010), $[H^+]$ the concentration of $H^+$
determined from pH = - log $[H^+]$, $K_W$ the dissociation constant of water (function of T, S and pressure),
$K_1$ and $K_2$ the first and second dissociation constants of carbonic acid (function of T, S and pressure,
Luecker et al., 2000). The estimate of atmospheric $CO_2$ includes site-specific offsets relative to
reconstructed surface water $CO_2$ to account for observed local disequilibrium (+21 ppm and +12 ppm
at ODP Sites 999 and 871, respectively).
**2.6 Uncertainty.**
*2.6.1    Analytical uncertainty.*
The uncertainty on the measured $\delta^{11}B$ is expressed as the external uncertainty which includes
instrumental error and chemical separation of the sample (see a detailed discussion in John and Adkins,
2010). This was determined empirically by long-term repeat measurements of JCp-1 subject to the same
chemical purification as our foraminiferal samples. As discussed by Rae et al. (2011) this uncertainty
is dependent on the intensity of the $^{11}B$ signal and is expressed here by the following relationship defined
during the duration of this study at the University of Southampton (Anagnostou et al., 2019), for $^{11}B$
intensities <0.54V:
$$2\sigma = 129600 \, e^{-212 \, x \, [^{11}B]} + 0.3385 \, e^{-1.544 \, x \, [^{11}B]} \quad (8).$$

where $[^{11}B]$ is the intensity of $^{11}B$ signal in volts. The $\delta^{11}B$ uncertainty for $^{11}B$ intensities > 0.54V is
0.15‰ (at 95% confidence).
*2.6.2    pH and $CO_2$ uncertainty.*
The $CO_2$ uncertainty we report was calculated with a Monte Carlo simulation (10, 000 realisations) in
order to fully account for the uncertainty in all variables used in the calculation of pH and $CO_2$ ($\sigma_{CO_2}$
$\delta11B\text{-derived}$). The shape of the uncertainty distribution sampled is either normally distributed (for
temperature, salinity and $\delta^{11}B$) or uniform (for alkalinity, as discussed above). The maximum
probability of all realisations was used as the central value for $CO_2$ and an error envelope at 1 and $2\sigma$
was calculated based on the 68% and 95-% distribution of the realisations.

*2.6.3    Uncertainty on the $CO_2$ offset*
To constrain the offset between $\delta^{11}B$-derived $CO_2$ and ice core $CO_2$, each sediment age is compared to
the ice core $CO_2$ record by interpolation of the record of highest resolution (in this case the $\delta^{11}B$ record
onto the ice core compilation). To fully account for age uncertainty when interpolating the sediment
age to the well-dated ice core record, a distribution of the ice core data was calculated within the $4\sigma$
uncertainty of the $\delta^{11}B$ age and weighed by the respective likelihood based on the age difference
between ice core and sediment core.
The $CO_2$ offset (or residual) is defined by:
$$Offset_{CO_2} = CO_{2 \, \delta11B\text{-derived}} - CO_{2 \, ice} \quad (9)$$



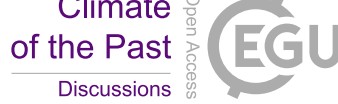

The uncertainty on this offset ($\sigma_{offset}$) accounts for the uncertainty of the interpolated ice core $CO_2$
($\sigma_{CO2.interpol}$) and the one of the $\delta^{11}B$-derived $CO_2$ ($\sigma_{CO2.\delta11B\text{-}derived}$), such as :

$$\sigma_{offset} = \sqrt{\sigma^2_{CO2.interpol} + \sigma^2_{CO2.\delta11B-derived}} \quad (10)$$

### 2.7 The relationship between $\delta^{11}B$ -derived pH and $\Delta F_{CO2}$.

The linear relationships between the relative $CO_2$ forcing $\Delta F_{CO2}$ and pH are determined with a York
regression (York et al., 2004) that accounts for the uncertainty in both the independent and dependent
variable (i.e. x and y axes). The ice core $CO_2$ interpolation used to calculate $\Delta F_{CO2}$ and uncertainty is
determined as described in section 2.6 (Hain et al., 2018).

### 2.8 Optimising the *G. ruber* $\delta^{11}B$ borate-foraminifera calibration.

An optimised *G. ruber* calibration was obtained by minimising the root mean square error (RMSE) of
the average offset between $\delta^{11}B$-derived $CO_2$ and ice core $CO_2$. The steps are illustrated in Figure S1.
In order to optimise the calibration, 10,000 simulations of $\delta^{11}B_{borate}$ and $\delta^{11}B_{foraminifera}$ from the calibration
of Henehan et al. (2013) were performed within their normally distributed uncertainty ($1\sigma$), from which
we defined the same number of linear models each including their slope and intercept. Then, we
calculate the equilibrium pH and resultant equilibrium $\delta^{11}B_{borate}$ from ice core $CO_2$ and the assumed
constant TA at each core site. The $\delta^{11}B_{borate}$ from the 10,000 linear models is then calculated and the
difference to the ice core-derived $\delta^{11}B_{borate}$ is determined. The linear model calibration that yields the
minimum RMSE between these two borate variables defines the new $\delta^{11}B_{borate\text{-}foram}$ calibration. Unless
indicated otherwise, the pH results presented in this study are calculated with the published calibration
(Henehan et al., 2013), and the results with the optimised calibration presented in section 4.2.6.

## 3    Results.

### 3.1 Temperature and fragment counts.

The SST at ODP Sites 999 and 871 show a cyclicity that agrees with the well-known glacial interglacial
cycles of the late Pleistocene (Figure 2). The Mg/Ca-SST corrected for pH (Figure S2) shows lower
temperatures of about 0.2 to 2.5°C, yet the glacial variation structure is maintained. The SST determined
from *G. ruber sl* Mg/Ca uncorrected (red filled circles, Figure 2B) at Site 871, show systematically
cooler temperatures than *G. ruber ss* (black filled circles). The fragmentation index (Figure 2) at ODP
871 range from 20 to 50 % and follow the well-documented "Pacific style" dissolution cycles (Sexton
and Barker, 2012) with well-preserved carbonate (low fragments) during glacials and less well-
preserved carbonates (higher fragments) during interglacials. The percentage sand typically
anticorrelates with fragmentation counts at both sites, although it is less clear at ODP 999, perhaps due
to the shorter record available. Fragmentation counts reach maxima at ODP 999 of 20 % during
interglacials and up to 50 % during marine isotope stage MIS 11 which is concomitant with the mid-
Brunhes dissolution interval (MBDI, Barker et al., 2006). The fragmentation counts at ODP 871 show
no substantive anomaly during the MBDI.





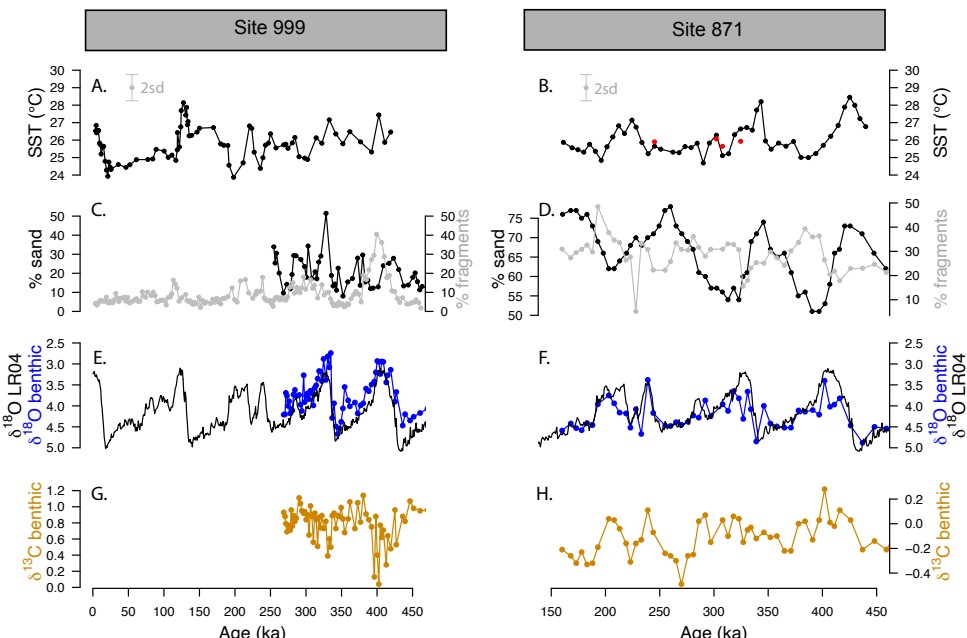

Figure 2. Mg/Ca derived temperature, coarse fraction (sand), fragmentation and benthic $\delta^{18}$O and $\delta^{13}$C at ODP sites 999 and 871. **A, B:** Temperature at ODP 999 (from *G. ruber ss*, black, Schmidt et al., 2006) and ODP 871 (*G. ruber ss*, black, *G.ruber sl*, red, 2sd indicated by the grey error bar). **C, D:** Fragmentation index (light grey, data from Schmidt et al. (2006) for ODP 999) and sand (black line). **E, F:** Benthic *C. wuellestorfi* $\delta^{18}$O (blue) and LR04 benthic $\delta^{18}$O stack (black). A correction of +0.48‰ is applied to our $\delta^{18}$O data in order to adjust for species offset between *C. wuellestorfi* and LR04. **G, H:** Benthic *C. wuellestorfi* $\delta^{13}$C (orange).

## 3.2 pH and CO₂ reconstructions.

The $\delta^{11}$B, pH and $\delta^{11}$B-derived absolute $CO_2$ (Figure 3) from Sites 871 and 999, show clear cyclicity related to glacial-interglacial cycles. The $CO_2$ values carry an average uncertainty of ±48 ppm and the mean offset from the ice core $CO_2$ for a combination of the two records is 4.6 ±49 (2σ) ppm showing that there is a minor overestimation of $CO_2$ using the boron method yet it agrees on average well within uncertainty. The RMSE of the $CO_2$ offset for the combined record is 25 ppm.

Despite the overall close agreement between $\delta^{11}$B-derived $CO_2$ and ice core-derived $CO_2$, each of our $\delta^{11}$B-$CO_2$ records exhibit some short-lived intervals where the offsets from the ice core record are larger. This is further revealed by the residual $CO_2$ and the identification of the data above the upper quartile (i.e. the upper 25% of the data, Figure S3). Those data do not appear to be randomly distributed and instead occur at ~100 ky, ~220-290 ky and ~390 ky at ODP Site 999, in all three cases during the early stages of the glaciation (except for the MIS 8 glacial at 280ky, Figure S3). The mismatches with the ice core at ODP Site 871 show a similar temporal pattern occurring at ~220 and ~300 and ~ 390 ky (i.e. at glacial inceptions).





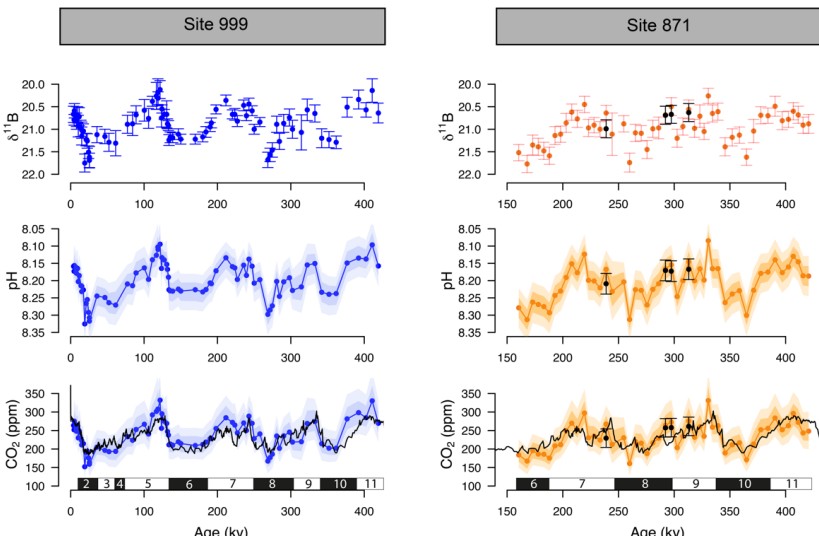

Figure 3. $\delta^{11}B$, pH and boron-derived $CO_2$ at site 999 and 871. $\delta^{11}B$ of *G. ruber ss* and *sl* (top row), boron-derived pH (middle row) and $CO_2$ (bottom row) reconstruction from two core locations: ODP 999 (blue, this study and published data, Foster, 2008; Henehan et al., 2013; Chalk et al., 2017) and ODP 871 (orange, this study). The black line in the $CO_2$ panels is the composite Antarctic ice core $CO_2$ record (Bereiter et al., 2015). All $\delta^{11}B$-derived data points are from *G. ruber ss* except black dots at ODP Site 871 measured on *G. ruber sl*. Numbers at the bottom of the $CO_2$ records represent marine isotope stages (Black box for glacials and white box for interglacials). Note the age scale is different at site 999 and 871.

### 3.3 Contrasting $\delta^{11}B$ between morphotypes.

Within error, the few measurements of $\delta^{11}B$ *G. ruber sl* at ODP 871 all agree with $\delta^{11}B$ *G. ruber ss* (Figure 3) albeit the $\delta^{11}B$ of *G. ruber sl* is higher than *G. ruber ss* for all 4 data pairs available. The $CO_2$ derived from *G. ruber sl* (Figure 3) is on average 15 ppm lower than the one derived from *G. ruber ss*; though the much lower resolution (n=4) impedes a thorough comparison at this stage. The $\delta^{18}O$ and $\delta^{13}C$ of both morphotypes were compared for the whole records at ODP 871 (Figure S4) and a cross-plot shows a moderate to good agreement between *G. ruber ss* and *sl* ($r^2$=0.55 and 0.22 for $\delta^{18}O$ and $\delta^{13}C$ respectively, Figure S5). This is in contrast to other studies (e.g. Wang et al., 2000; Steinke et al., 2005) that show $\delta^{18}O$ in *G. ruber sl* to be systematically higher.

### 3.4 Dissolution experiments.

The leaching experiments on *T. sacculifer* and *G. ruber ss* show a different response for the two species (Figure S6). While *G. ruber ss* show no significant variation in measured $\delta^{11}B$ under different treatments, *T. sacculifer* shows no systematic variations in $\delta^{11}B$ for the control and first two treatments (leached in 2 and 4 hours in 0.0001M $HNO_3$, pH 4) but shows a ~1‰ shift (relative to the control) towards lighter $\delta^{11}B$ after 6 hours at pH 4.





### 3.5 Relationship between $\delta^{11}$B-pH and $CO_2$ forcing from the ice core.

A cross plot of $\delta^{11}$B-derived pH $CO_2$ forcing from the ice core record for each of our marine core study sites is shown in Figure 4 and is compared to the theoretically-derived approximate $\Delta FCO_2/\Delta pH$ relationships as adopted by Hain et al. (2018): -1:1 W/m$^2$ (dashed black line); $CaCO_3$ addition/removal (-0.9:1 W/m$^2$ plain yellow line); DIC addition/removal (-1.3:1 W/m$^2$ dotted-dashed blue); and warming/cooling temperature forcing (-1.1:1 W/m$^2$, dashed red). Our analysis includes full propagation of uncertainty in pH, in contrast to Hain et al. (2018) who considered only the reported uncertainty of $\delta^{11}$B$_{borate}$ in their validation exercise. In both cases the uncertainty in $\Delta F_{CO2}$ accounts for the error in interpolation arising when comparing age-uncertain $\delta^{11}$B-derived pH with $\Delta F_{CO2}$ from the well-dated and high-resolution ice core $CO_2$ record (see methods 2.7 and 2.6 for details). This treatment of $\Delta F_{CO2}$ uncertainty is dominated by the spread of ice core $CO_2$ data points within the $\delta^{11}$B age uncertainty. The data are fitted with a York-type regression (thin black line; York et al., 2004) where the grey envelope represents the uncertainty of the linear relationship that best represents the data (i.e., the envelope is not the prediction interval), considering the uncertainty in pH and $\Delta F_{CO2}$. The regressed slope is $\Delta F/\Delta pH$ = - $15.42 \pm 0.8$ W/m$^2$ (-1.2:1 relative to basic formalism) and shows a good agreement with the theoretical temperature and DIC driven relationships.

The effect of the uncertainty assigned to pH (fully propagated or using the measurement uncertainty of the boron isotope) on the regressed slope is shown in Figure S7. The slope of the York regression when using the uncertainty from $\delta^{11}$B only, as in Hain et al. (2018), shows a very close agreement with the basic formalism, with a slope of $\Delta F/\Delta pH$= - $12.4 \pm 0.3$ W/m$^2$, (-1:1 relative to the basic formalism) but with a unsatisfactory goodness of fit (mean square weighted deviation, mswd) of 6, whereas propagating the full pH uncertainty based on our iterative Monte-Carlo simulations improves goodness of fit to ~1 at a $\Delta \log_{10}CO_2/\Delta pH$ of -1.2:1 (Figure 4).

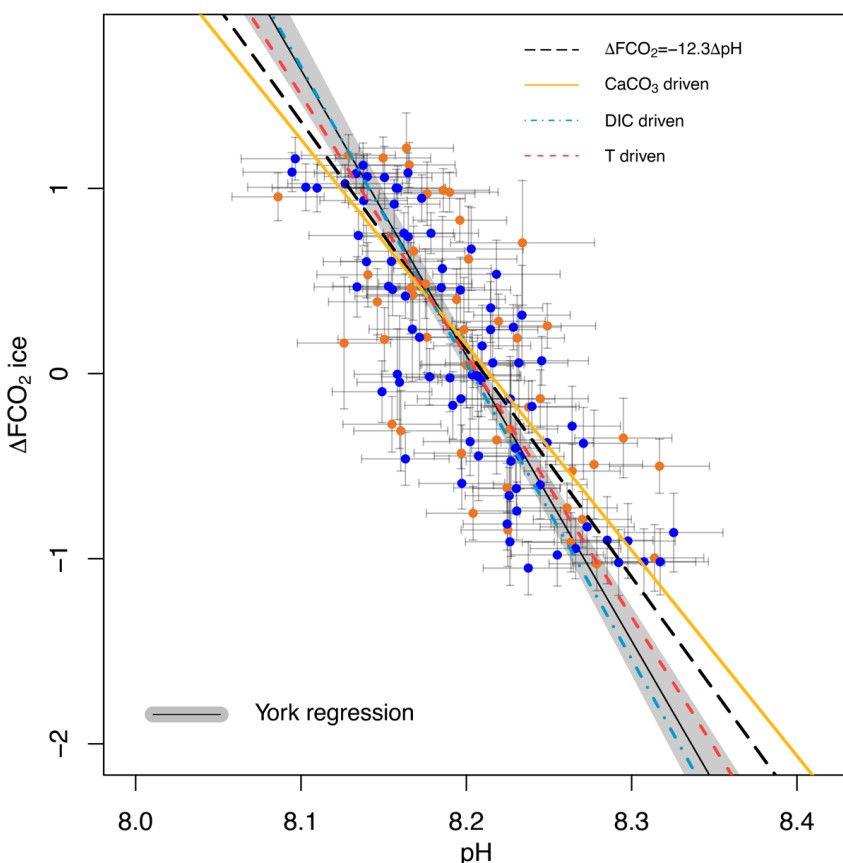

Figure 4. Ice core based $\Delta FCO_2$ ($CO_2$ forcing) vs. $\delta^{11}B$-based pH for ODP 999 (blue filled circles, this study and
published data from Foster, 2008; Henehan et al., 2013; Chalk et al., 2017) and 871 (orange filled circles). The
lines show the relationship between $\Delta FCO_2$ and pH for the simplified formalism (see method) $\Delta FCO_2= -12.3\Delta pH$
(black dashed line), and when driven by changes in DIC only (blue, $\Delta F/\Delta pH= -16$ W/m$^2$), $CaCO_3$ (yellow,
$\Delta F/\Delta pH= -11.1$ W/m$^2$) and temperature T (red, $\Delta F/\Delta pH= -14.1$ W/m$^2$). The York regressed line (black line and
grey shade) falls between the theoretical only pH-driven line (black) and $CaCO_3$ line (yellow).
**4   Discussion.**
**4.1 Cyclicity in foraminifera preservation.**
Percentage fragments and sand fraction ($> 63\mu m$) at both studied core sites are anticorrelated and show
a clear cyclicity, with better preservation of carbonates during glacial periods (Figure 2). The
anticorrelation is clearer at ODP 871 where we have the longest record (Figure 2).  Sexton and Barker
(2012) suggest that this Pacific Ocean pattern of preservation (Farrell and Prell, 1989) initiated after the
mid Pleistocene transition (MPT) around 1 Ma, and that preservation cycles in the Pacific prior to MPT
showed a more "Atlantic style" of dissolution with better (poorer) preservation occurring during
interglacials (glacials). Several data sets (deep oxygen and carbon isotopes, carbonate ion data, and
sortable silt) point towards a strengthening of ventilated deep Pacific waters (lower circumpolar deep
water LCDW) that lead to the better preservation during glacials in the Pacific after the MPT (Sexton
and Barker, 2012).

The observation that the fragmentation records of sites 999 and 871 covary is likely attributable to the
different water masses that fill the Caribbean basin relative to the rest of the Atlantic basin. During
glacials, the deep Atlantic is filled by nutrient- and carbon-rich corrosive southern sourced waters
(Antarctic Bottom Water, AABW) with a reduced contribution from the less corrosive, nutrient-poor
North Atlantic Deep Water (NADW) (Oppo and Lehman, 1993) causing calcareous sediments in the
deep Atlantic Ocean >2500 m to be less well-preserved during glacials than interglacials. The opposite
pattern of dissolution is seen in the Caribbean because shoaling of the northern sourced waters during
glacials produces a mid-depth well-ventilated water mass that feeds into the Caribbean through its
deepest sill (~1900 m, Johns et al., 2002). Thus the deep Caribbean is filled with less corrosive waters
during glacials than interglacials improving the preservation of carbonate during glacials in a similar
pattern to a Pacific styled dissolution cycle albeit in response to Atlantic circulation changes. During
interglacials, the Northern sourced waters are mixed with corrosive southern sourced waters (Antarctic
Intermediate Waters and upper circumpolar deep waters) leading to less well-preserved sediments.
**4.2 Causes of offset between $\delta^{11}B$ –derived and ice core $CO_2$.**
The $\delta^{11}B$-derived $CO_2$ record from both of our study sites is in very good agreement with the ice core
record, with an average offset for combined both cores of 4.6 ±49 (2σ) ppm and corresponding RMSE
of 24.7 ppm. The $CO_2$ offset calculated with Mg/Ca-SST corrected for pH is shown in Figure S8 for
comparison and the average is -4.8 ±42 (2σ) ppm, showing a reduced offset of 9 ppm compared to
treatment with no pH correction on SST (a difference of -11 ±14 (2σ) and -8 ±12 (2σ) ppm at ODP
site 871 and 999, respectively). This difference is due to the pH correction lowering the SST estimates
on average without greatly changing the temporal structure of pH and $CO_2$ offsets.
In both treatments, the RMSE is smaller than the average $CO_2$ uncertainty of ±48 ppm (2σ, 95%
confidence) for each datapoint. However, the minor $CO_2$ offsets observed in both records do not appear
to be random and tend to fall during the first half of each glacial cycle (Figure S2). In order to have the
highest confidence in $CO_2$ reconstructions using $\delta^{11}B$, this pattern warrants further investigation (see
below). We only discuss the $CO_2$ records calculated without a pH correction on SST.
*4.2.1 Comparison between morphotypes of G. ruber*
If as others suggested (e.g. Wang et al., 2000; Steinke et al., 2005; Numberger et al., 2009) *G. ruber sl*
and *G. ruber ss* occupied different depth habitats, then inadvertent sampling of the cryptic *G. ruber sl*
morphotype might conceivably produce the biases we observe between $\delta^{11}B$-derived $CO_2$ and
atmospheric $CO_2$ from the ice cores. However, while our Mg/Ca-derived temperatures for *G. ruber sl*
and *G. ruber ss* display variable offsets, they are within uncertainty (Figure 2) and our $\delta^{18}O$ and $\delta^{13}C$
data for the two morphotypes at ODP 871 show a good agreement with no consistent differences (Figure
S4). Thus, while the water column profile of $\delta^{18}O$ and $\delta^{13}C$ can be affected by factors other than
temperature, salinity and biological productivity (e.g, carbonate ion effect, Spero et al., 1997), overall,
our data suggest that the two morphotypes we analysed shared similar depth habitat preferences.
Henehan et al. (2013) found that *G. ruber ss* and *sl* record similar $\delta^{11}B$ in core-top sediments, and
through necessity, used mixed morphotypes in their culture study. The $\delta^{11}B$-derived pH and $CO_2$ for *G.*
*ruber sl* examined here are consistently higher and lower, than *G. ruber ss* by around 0.05 pH units and
15 ppm $CO_2$, respectively (Figure 3). This is contrary to expectation if *G. ruber sl* lived in deeper more
acidic waters as suggested by other studies (Wang et al., 2000; Steinke et al., 2005), but consistent with
some data sets that show that the habitat of *G. ruber ss* and *sl* can vary by location (Numberger et al.,
2009). We acknowledge that the scarcity of *G. ruber sl* in our samples means that our data set for this
morphotype is too small to draw firm conclusions and this warrants further investigation at other study
sites. Nonetheless, the closeness of the morphotypes in terms of $\delta^{11}B$ and depth habitat throughout our
record implies any inadvertent sampling of *G. ruber sl* in the *G. ruber ss* fraction in this study and
location would not significantly bias our reconstructions.





*4.2.2   Change in upwelling and CO$_2$ disequilibrium.*
ODP sites 871 and 999 are both located today in stratified oligotrophic environments with a deep
modern thermocline (base of the thermocline is at ~ 200 and 400 m at ODP 871 and 999, respectively;
Olsen et al., 2016). It should be noted, however, that both sites are situated relatively close to regions
displaying ΔpCO$_2$ >40 ppm (Figure 1).  However, if local upwelling occurred over the study interval,
or if these areas of upwelled water expanded, we would expect these periods to be characterised by
relatively low SST, high surface δ$^{18}$O, and low surface δ$^{13}$C due to an increased influence of deep colder
and more remineralised waters. The identified anomalous intervals in residual CO$_2$ at ODP 871 (e.g at
~210, ~290 ky, Figure 5) show no particular anomaly in planktonic C and O isotopes (Figure S4) or in
SST (Figure 2, Figure S9), ruling out significant variations in upwelling at that site. Equally, no SST
anomaly was identified at ODP 999 to be coincident with the intervals of high residual CO$_2$ (Figure S9).
This suggests the CO$_2$ anomalies revealed in Figure 5 are not the result of enhanced local disequilibrium
via sub-surface water mixing.
*4.2.3   Partial dissolution.*
The CO$_2$ derived from *G. ruber* δ$^{11}$B at ODP 999 and 871 appears to show, at first order at least, positive
CO$_2$ offset during periods of high fragmentation (~100, ~210, ~400ky,  red filled circles in Figure 5,
defined by the upper 25% quantile of fragments) following a "Pacific style" dissolution cycle (better
preservation and lower fragmentation during glacial periods). Periods of high fragmentation at ODP
site 999 and 871 both (incidentally) correspond to a positive CO$_2$ offset 75% of the time, and 25% to a
negative CO$_2$ offset, (note that values close to 0 were omitted in this calculation). We also note that
almost all CO$_2$ offsets uncertainty (2σ) overlap with the 0 line, hence the percentage of CO$_2$ offset that
are above or below the 0 line should be interpreted with caution.

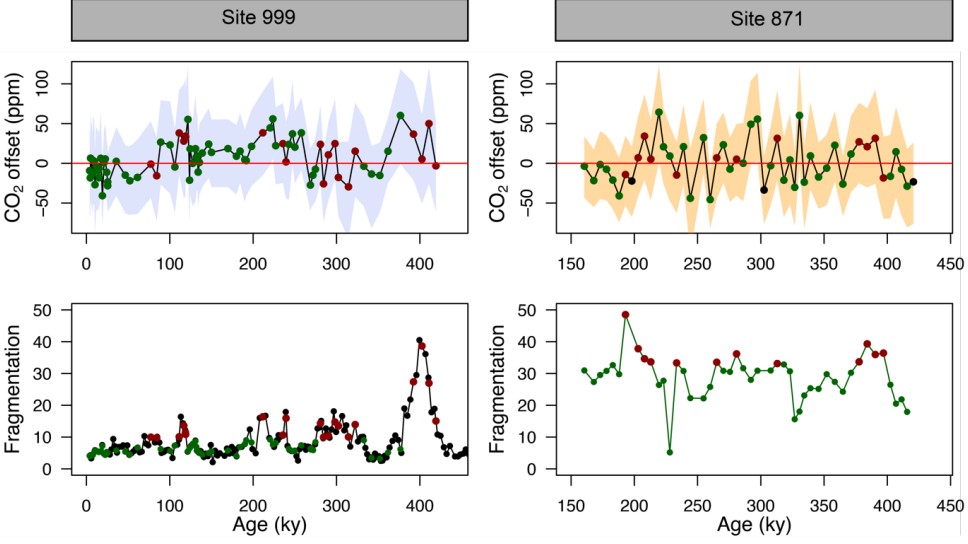

Figure 5. Top panels: CO$_2$ offset (defined as offset = CO$_{2\_δ11B\text{-derived}}$ − CO$_{2\_ice}$) for ODP Sites 999 (this study and
Chalk et al., 2017) and 871. See text for error bars calculations. Bottom panels: fragmentation index at Site 999
(Schmidt et al., 2006) and 871 (this study). Red dots in the lower panels are the fragments above the upper quartile
(and corresponding CO$_2$ in the upper panel, red dots). Green dots represent periods of low fragments below the
upper quartile (and corresponding CO$_2$ in the upper panel, green dots).



In detail however, a cross-plot of fragment counts and $CO_2$ offset (Supplementary Figure S10) fitted
with a linear regression shows no significant correlation for both core site 999 ($r^2$=0.07, p=0.02) and
871 ($r^2$=0.01, p=0.62). Although it should be noted that this simple linear regression presupposes a
linear relationship between the variables and does not account for the significant uncertainty in both
$CO_2$ offset and fragmentation index. In particular, the $CO_2$ offset carries the uncertainty from the
interpolated ice core $CO_2$ (see methods). Fragment counts at ODP 999 also come with the additional
uncertainty related to the interpolation of the record of Schmidt et al. (2006), whereas fragments counts
and $\delta^{11}B$-derived $CO_2$ at 871 are measured on the same samples. A cross-correlation function also shows
no correlation between $CO_2$ offset and fragmentation (Figure S11).
While it seems unlikely the small offsets observed are fully explained by partial dissolution, the positive
$CO_2$ offsets observed during some periods of high fragmentation index (Figure 5), are in line with the
trend observed in *T. sacculifer* during our dissolution experiments that showed a decreased $\delta^{11}B$ (that
translates to higher $CO_2$) with progressive dissolution (supplementary Figure S6). However our
dissolution tests, are consistent with field studies (e.g. Seki et al., 2010), and suggest that *G. ruber* $\delta^{11}B$
is relatively robust to dissolution (see section 3.4 above). The pattern observed here for *T. sacculifer*
has been documented in other studies where lower $\delta^{11}B$ is observed for core-top samples from deeper
ocean sites bathed by waters with low calcite saturation state (Hönisch and Hemming, 2004, Seki et al.,
2010). Tests of *T. sacculifer* can contain a significant proportion of gametogenic calcite (ranging 30 to
75% of the weight of pregametogenic calcite , Bé, 1980; Caron et al 1990) which forms at the end of
the life cycle in deeper lower pH cold waters. It has been suggested that $\delta^{11}B$ is lower in gametogenic
calcite than in the primary test (Ni et al., 2007) reflecting the digestion and expulsion of symbionts (Bé
et al., 1983) before gametogenesis, driving a relative acidification of the micro-environment (no $CO_2$
uptake by photosynthesis) around the foraminifera (Zeebe et al. 2003; Hönisch et al,. 2003; Henehan et
al. 2016), and movement to deeper more acidic waters during that life-stage. It has also been shown that
this gametogenic calcite is more resistant to dissolution (Hemleben et al., 1989; Wycech et al., 2018)
resulting in partial dissolution acting preferentially on ontogenic calcite driving $\delta^{11}B$ in the residual test
to lower isotopic composition.
While the decrease in $\delta^{11}B$ in dissolved test of *T. sacculifer* is well explained by the lighter isotopic
composition of gametogenic calcite, *G. ruber* tests do not contain such gametogenic calcite (Caron et
al., 1990). Hence, if the observed occasional decrease in $\delta^{11}B$ (low pH, high $CO_2$) was caused by partial
dissolution, it needs to be explained by other processes. It should also be considered that the dissolution
experiments performed here could be of longer duration (e.g. Caron et al., 1990; Sadekov et al., 2010)
and be more quantitative (e.g. with alkalinity of leaching acid, trace element data and foraminifera
weight data to evaluate the degree of dissolution over time). Furthermore, alternative measures and
proxies of dissolution may yield more quantitative constraints (e.g. benthic B/Ca as an indicator of
bottom water carbonate ion concentration) on the importance of dissolution in generating our observed
$CO_2$ offsets.
Some studies have shown that laboratory dissolved specimens of *T. sacculifer* (Sadekov et al., 2010)
and naturally dissolved specimens of *G. ruber* (Iwasaki et al., 2019) undergo targeted partial preferential
dissolution of the shell. However, variations in intra-shell $\delta^{11}B$ are currently unknown due to limitations
in laser ablation techniques that impede a direct evaluation of $\delta^{11}B$ heterogeneity in foraminifera
chambers. Future studies are needed to constrain the $\delta^{11}B$ spatial distribution in foraminiferal shells
caused by potential variations in $\delta^{11}B$ from dissolution, ontogeny (e.g. Meilland et al., 2021) and/or
vital effects (e.g. change in photosymbiotic activity throughout the life cycle, Lombard et al., 2009,
Henehan et al., 2013, Takagi et al., 2019).
In the absence of these constraints, and given the limitations of our dissolution experiments, we
conclude that partial dissolution is unlikely to be a significant driver of the $\delta^{11}B$-$CO_2$ records we present
here. Even though it was thought to be a species susceptible to dissolution (Berger, 1970), we confirm
that the $\delta^{11}B$ of *G. ruber* appears more resistant to dissolution-driven modification than *T. sacculifer*.



### 4.2.4. *Effect of dissolution on Mg/Ca and calculated CO₂.*

The direction of change of Mg/Ca with partial dissolution is towards lower ratios in partially dissolved foraminifera (e.g. Brown and Elderfield, 1996; Dekens et al., 2002; Fehrenbacher and Martin, 2014). If the Mg/Ca is impacted during periods of high fragmentation, the lower ratio would result in lower temperatures leading to lower calculated $CO_2$ values (equation 7). This effect is opposite to the occasional positive deviation of $CO_2$ observed during intervals of high fragmentation at ODP Site 999. While the weak correlation between fragmentation and $CO_2$ precludes a firm interpretation of dissolution effect, we conclude that the effect of partial dissolution on Mg/Ca ratio and resulting $CO_2$ (if any) is negligeable and not responsible for the $CO_2$ offsets observed during intervals of high fragmentation.

### 4.2.5. *Change in the second carbonate parameter, alkalinity.*

Past changes in TA are poorly constrained, although some constraints are starting to emerge for the late Quaternary (e.g. Cartapanis et al., 2018). However, since pH is directly determined by $\delta^{11}B$, pH defines the ratio of alkalinity to DIC (see supplementary information S12). Hence, at any given pH, any change in alkalinity must be counteracted by a change in DIC, which has the opposing effect on $CO_2$. This is demonstrated by the tight relation between pH and $CO_2$ highlighted by our data (Figure 4). The largest residual $CO_2$ is ~50 ppm at ODP 999. To produce an effective alkalinity-driven change in $CO_2$ of this magnitude at a given pH requires an alkalinity reduction of about ~300 to 500 μmol/mol (supplementary Figure S13). This is far larger than any expected change over a glacial cycle (Cartapanis et al., 2018, Hönisch et al., 2009). We therefore rule out varying TA as the cause of the minor $CO_2$ offsets observed (Figure 5).

### 4.2.6 *Improving the $\delta^{11}B$ -pH G. ruber calibration*

A further potential cause for the minor offsets observed between $\delta^{11}B$-derived and ice core $CO_2$ could be a small inaccuracy in the calibration between $\delta^{11}B$ of foraminifera and borate for *G. ruber* (Henehan et al., 2013). Having the ice core data to compare with $\delta^{11}B$-derived $CO_2$ offers an opportunity to explore the effect of altering the input variables of the pH-$CO_2$ calculation to see if doing so improves the fit to ice-core values. Note that such an exercise is for illustrative purposes only because we seek to retain the independence offered by the $\delta^{11}B$-calibrated data in the context of $CO_2$ forcing (section 4.3). Nonetheless, in future work we suggest this calibration can be applied in tandem to the empirical relationship of Henehan et al. (2013). The published (Henehan et al., 2013) and obtained optimised calibration (Figure S14) are:

$$\delta^{11}B_{borate} = \frac{\delta^{11}B_{foram} - 8.87(\pm 1.52)}{0.60(\pm 0.09)} \quad \text{(Henehan et al., 2013)}$$

$$\delta^{11}B_{borate} = \frac{\delta^{11}B_{foram} - 6.46}{0.72} \quad \text{(optimised calibration)}$$

The newly calculated $CO_2$ with the updated calibration shows an improved average $CO_2$ offset (Figure 6) of -3.43 ± 41 (2σ) ppm (vs 4.6 ± 49 (2σ) ppm with the calibration of Henehan et al., 2013) and an RMSE of 20.68 ppm (vs. 25 ppm with the published calibration).

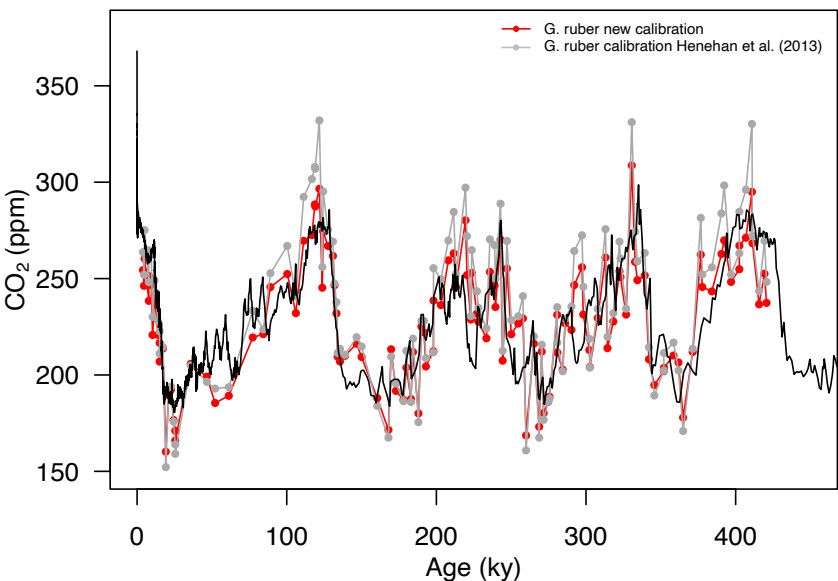

600

Figure 6. Composite $\delta^{11}$B-derived $CO_2$ from both core sites 999 and 871 using the published $\delta^{11}B_{borate\text{-}foram}$
calibration (grey points, Henehan et al., 2013) and the improved calibration (red points). The black line is the
Antarctic composite ice core $CO_2$ record (Bereiter et al., 2015).

When analysing the $CO_2$ offset using the optimised *G. ruber* calibration and the fragmentation index at
each core location (same approach as Figure 5), we observe that intervals of high fragments (defined as
values above the upper quartile) are no longer preferentially associated with positive $CO_2$ offset (Figure
S15). Intervals of high fragments at site 999 have 50% chance of corresponding $CO_2$ with positive
offsets to the ice core (and 50% with negative offset to the ice core). Intervals of high fragments at site
871 have 56% of corresponding $CO_2$ with a positive offset to the ice core (and 44% with a negative
offset to the ice core). This analysis shows that a small change in the borate *G. ruber* $\delta^{11}$B calibration
does not cause any visual correlation between $CO_2$ offset and fragmentation index (Figure S15), and
that uncertainty in the $\delta^{11}B_{foram\text{-}borate}$ calibration of Henehan et al. (2013) can – at least partly – explain
the minor discrepancies we observe between $\delta^{11}$B-derived and ice core $CO_2$.

### 4.3 Relative $CO_2$ forcing and pH.

Our new pH data, added to the existing compilation, show a good agreement with the formalism defined
by Hain et al. (2018; Figure 4). It should be noted that $CO_2$ in this case is provided by the ice core
directly and is not estimated from the $\delta^{11}$B-derived pH. As discussed above, because these two proxies
are independent of one another, the slope of their relationship may be used to interrogate the
mechanisms of $CO_2$ change. Our data fall between the $CaCO_3$ (yellow plain line) and the DIC (dotted-
dashed blue line) end-members suggesting that the $CO_2$ change observed on glacial-interglacial
timescales was driven by a mix of mechanisms rather than to a single cause. This is in line with studies
that require a number of mechanisms to explain glacial interglacials $CO_2$ change (soft tissue pump,
carbonate compensation pump and thermal pump, e.g. Brovkin et al., 2007, Kohfeld and Ridgwell,
2009, Hain et al., 2010, Chalk et al., 2019, Sigman et al., 2021). We note that this is a preliminary
interpretation because of the sensitivity of our finding to pH uncertainty (section 3.5, Figure S7). To
overcome this ambiguity in estimating past $\Delta FCO_2$ and to better deconvolve the driving mechanisms of
glacial/interglacial $CO_2$ change, we recommend that future studies collect pH data at higher temporal





resolution to examine the change in slope through a glacial cycle and strive to further quantify and
reduce uncertainties related to pH determination.
The close agreement of the pH and ice core $CO_2$ data with the theoretical relationships has a number of
consequences for the reconstruction of $CO_2$ change during periods of Earth history beyond the ice core
$CO_2$ and climate records where constrains on $\delta^{11}B_{sw}$ and the second carbonate parameter and
temperature are uncertain. The $\Delta pH$ formalism still requires an estimation of $\delta^{11}B_{SW}$ and temperature
(for the $pK_B$ term, equation 5) however, as discussed in Hain et al. (2018), while absolute reconstruction
of pH is significantly influenced by estimates of $\delta^{11}B_{sw}$ and temperature, reconstruction of relative pH
change ($\Delta pH$) is inherently much less sensitive to these input variables.
Reconstructing $\Delta F_{CO2}$ from $\Delta pH$ is ideally applicable only on relatively short timescales less than 1
Myrs, when $\delta^{11}B_{sw}$ is likely to be constant given the multi-million year residence time of boron in the
ocean (Lemarchand et al., 2000, Greenop et al., 2017). Furthermore, to reconstruct $\Delta F_{CO2}$ (and thus
climate sensitivity to $CO_2$), the formalism can be applied as long as, in equation 2, $\Delta pH$ remains the
overwhelming control. This is dependent on the residence time of carbon in the ocean with respect to
silicate weathering – approximately one million years (Hain et al., 2018) such that net carbon addition
to or removal from the Earth System through volcanic outgassing or silicate weathering is likely to be
minor over the million-year timescale. However, during some short events, such as for instance the
Palaeocene-Eocene Thermal Maximum, considerable carbon was added to the system in <200 kyr (e.g.
Gutjahr et al., 2017) invalidating the formulation described in equation 2 on these intervals. We also
emphasize that this formalism is only valid as long as core sites remain in equilibrium with the
atmosphere.

## 4.4 Caveats and future studies.

The aim of this study is to evaluate the capacity of the $\delta^{11}B$-pH proxy in *G. ruber* to accurately
reconstruct atmospheric $CO_2$ in the past. The overall agreement with the high confidence ice core $CO_2$
(e.g. Bereiter et al., 2015) is very promising and gives confidence to $\delta^{11}B$-derived $CO_2$ reconstructions
beyond the ice core record (>800 ky). We have however identified occasional, minor offsets between
the two records and explored potential drivers (partial dissolution, $\delta^{11}B$ borate-foram calibration, local
air-sea disequilibrium). It is likely that the minor disagreement observed (Figure 5) has a combination
of drivers and that a single mechanism is not solely responsible for the $CO_2$ offsets observed. To confirm
these trends, we recommend future work to focus on the following:
(1) The improved $\delta^{11}B$ calibration approach should be tested at more core locations. We note that the
improved calibration to the ice core records reported here was achieved using data from two sites. While
care is taken in the choice of study site to minimize air-sea $CO_2$ disequilibrium and sediment dissolution,
the newly defined improved $\delta^{11}B_{borate-foram}$ calibration should be seen as an exercise that is tailored to the
available data in this study, and future high-resolution studies can apply the method used here (section
4.4.5) to further test how the *G. ruber* calibration changes if $CO_2$ offsets occur in a similar fashion (i.e
at a particular time in each glacial cycles). We note the importance of high resolution (at least 3 ky)
sampling in future studies because most $CO_2$ offsets observed are short lived.
(2) A multiproxy approach is ideally needed. In particular, reliable indicators of temperature and
productivity, to assess change in upwelling and foraminifera ecology. We encourage future studies to
expand high resolution boron-derived $CO_2$ record and ancillary data (C and O isotopes, proxy of
carbonate preservation and bottom water corrosiveness, biological productivity) to further constrain the
capacity of the boron isotope pH/$CO_2$ proxy to generate reliable $CO_2$ records. As more recent IODP
expeditions include porewater data, constraints on bottom water conditions and degree of corrosiveness
at a given site will become available to evaluate the impact on $\delta^{11}B$ signals in foraminifera.



(3) Efforts should continue to decrease the analytical uncertainty associated with a $\delta^{11}B$ measurement
by MC-ICPMS because this still accounts for ~40% of the total uncertainty associated with each $\delta^{11}B$-
derived $CO_2$ estimate.
(4) We find little evidence to suggest that partial dissolution of foraminiferal tests (*G. ruber*) is a major
driver of uncertainty in $\delta^{11}B$–derived $CO_2$ estimates but more thorough experiments are desirable
because of site-to-site differences in foraminifera taphonomy.
**5. Conclusion.**
We carried out the most thorough test to date of the $\delta^{11}B$–pH ($CO_2$) proxy by comparing new high-
resolution (3 to 6 ky per sample) boron isotope–based pH and $CO_2$ at two locations with $CO_2$ from the
ice core record. Results suggest that the boron isotope proxy is robust and suited to reconstructing $CO_2$
to a precision of ±48 ppm (2σ, RMSE =25ppm) over this interval, with little or no systematic bias
shown by a mean residual of 4.6 ± 49 (2σ) ppm. This provides high confidence to the application of
the proxy beyond the reach of the ice core records.
Despite the overall good agreement, there are some minor short-lived $CO_2$ offsets that appear to have
some temporal structure and we explored a number of possible drivers. A visual correlation between
$CO_2$ offset and fragmentation index at core site 999 is observed (Figure 5) but is not statistically
significant. The effect of partial dissolution on $\delta^{11}B$ in *G. ruber* appears to be negligeable in our record,
but the possible heterogeneity of $\delta^{11}B$ within shells as well as variable susceptibly to dissolution of the
different parts of the foraminifera, encourages further exploration.
An revised $\delta^{11}B$ borate–foram calibration was calculated by minimising the offset between $\delta^{11}B$–
derived $CO_2$ and ice core $CO_2$ using published calibration (Henehan et al., 2013). While the new
calibration improves the fit to the ice core records, we caution against its use to estimate $CO_2$ given that
it is no longer independent of the ice core or the assumptions we make here to calculate $CO_2$ (i.e. that
TA is constant).
The formalism established by Hain et al. (2018) is robust, showing that relative $CO_2$ forcing in the past
can be determined from pH change alone, even in the face of significant uncertainty in $\delta^{11}B$ of seawater
and without the need to determine a second carbonate parameter. This will not only be of great interest
to determine $CO_2$ forcing in ancient geological times where $\delta^{11}B$ of seawater and a second carbonate
parameter are poorly constrained, but the nature of the observed relationship over the last 400 kyr
confirms that multiple drivers are likely responsible for glacial-interglacial $CO_2$ change.
**6. Data availability.**
All raw data will be provided as supplementary information once the manuscript is accepted.
**7. Author contribution.**
E.d.l.V generated boron isotope and elemental data and wrote the manuscript. E.d.l.V, T.B.C, M.P.H
and G.L.F analysed the data. G.L.F, T.B.C, M.P.H and P.A.W contributed to the editing and
reviewing of the manuscript. M.W, R.G and D.C generated oxygen and carbon isotope data and
fragmentation index data. R.G and D.C were supervised by T.B.C and G.L.F. C.L assisted with
foraminifera picking and boron isotope analysis. E.d.l.V, T.B.C and G.L.F designed the research.
**8. Competing interest.**
The authors declare they have no conflict of interest.
**9. Acknowledgment.**
We thank J. Andy Milton for assistance in MC-ICPMS and ICPMS analysis, and members of the "B-
team", Agnes Michalik and Matthew Cooper for clean laboratory assistance. This work was funded



by NERC grant NE/P011381/1 to GLF, PAW, TBC and MPH and by Royal Society Wolfson Awards
to both GLF and PAW.

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

as a paleo-pH indicator: Evidence from modeling." Paleoceanography **18**(2).

