# Peer review of "Orbital CO2 reconstruction using boron isotopes during 1 the late Pleistocene, an assessment of accuracy. 2"

_Climate of the Past, 2022_

## Referee Comment (RC2)

**Orbital CO$_2$ reconstruction using boron isotopes during the late Pleistocene, an assessment of accuracy – de la Vega et al**

review by William Gray (william.gray@lsce.ipsl.fr)

Boron isotopes in planktic foraminifera are a widely used proxy for reconstructing ancient CO2. As other studies have done previously, de la Vega et al use ice core CO2 as a test of accuracy; to this end, they present new G-IG boron isotope data, including data from a new sediment core site in a location close to equilibrium with the atmosphere today. This is only the second record of its kind (i.e. a record from a site close to equilibrium and over multiple glacial cycles) and is a very welcome addition as it helps overcome the assumption that a single site (i.e. ODP 999) has remained in equilibrium with the atmosphere. They go on to assess the accuracy with which CO2 can be reconstructed, and possible causes of the (albeit relatively minor) discrepancies (dissolution, second carb system parameter, calcite-borate calibration). Overall, I found the manuscript to be clear, well structured, and well-reasoned.

I have the following suggestions:

I think a greater exploration/discussion of the thermal influence on the carbonate/borate system (via the dissociation constants) in paleo CO2 reconstruction is warranted – basically, how sensitive is CO2 to accurate SST reconstruction? More sensitivity tests could be implemented - including keeping temperature constant throughout. Overall, I think further exploration/discussion of the thermal effects are warranted as, my guess is, this could be an important source of bias/uncertainty and it's helpful to understand what we need to improve. The authors mention they also use our iterative pH correction, and briefly mention this in the text, but I think it warrants further discussion; there is very good evidence from culture studies that pH influences Mg/Ca, and this is an influence we know is going to covary with atmospheric CO2.

The authors mainly rely on the calibration of Anand et al 2003 to derive Mg/Ca SSTs to use in their CO2 reconstruction. Although widely applied I don't think the calibration of Anand et al accurately describes the relationship between Mg/Ca and temperature (see figures 5 and 6 in Gray et al 2018). This is very apparent if you compare the measured CTD temperatures at the sediment trap site used by Anand, and the temperature calculated using Anand's own Mg/Ca data and their calibration line (see figure below). Using the calibration of Anand et the Mg/Ca are almost always too warm, and the seasonal cycle is about half of what it should be as winter temperatures are 4 degrees too warm (there is no way to explain this by 'sampling issues' as the Mg/Ca SSTs are warmer than any individual CTD measurement ever taken at the site in winter, and this is BATS so it's about the most heavily sampled place in the ocean). Basically, if you use Anand et al on their own data, you get the wrong answer. If the authors want to use Anand et al, I think there needs to be more justification (and 'it gives a better fit to the CO2 data/the core top SSTs' isn't a great reason).

[Figure]

*Figure above shows a comparison of Mg/Ca and CTD temperatures at the Sargasso Sea sediment trap site (the site used by Anand to derive their Mg/Ca calibration). I'm showing a LOESS fit to the Mg/Ca data (colored lines), rather than the individual data points to make it legible. The grey dots show all the individual CTD measurements ever taken at this site. The black line/grey shaded area is the expected temperature (and 95% CI) at the habitat depth of G. ruber.*

For the leaching experiment is T. sacculifer with or without sacc? This is really important as the lines of argument regarding the differential dissolution of the gametogenic calcite versus the rest of test only hold if it has a sacc (note there are similar arguments for a d18O dissolution effect in this species). The authors mention they didn't assess weight loss (a shame), but are there really no TE data for these samples? Was an aliquot not analysed as part of the boron isotope analysis to check for cleaning etc? If TE data are available it would be really informative to show how Mg changes with the leaching, as having some metric to be able compare to real world samples would make these leaching experiments much more useful.

For the uncertainty and the discussion of changing DpCO2 its worth noting that Earth System Models under glacial forcings typically simulate very small changes in DpCO2 – this can be seen in the figures below which show the pH difference minus the mean ocean pH difference in the IPSL model between PI and LGM forcings (taken from Gray and Evans 2019) – basically most of the ocean just reflects the change in atmospheric CO2 (95% range within ±0.05 units, equivalent to about ±40 uatm DpCO2). There is almost no residual pH change at the two sites used in the present study. I think this is really encouraging for paleo CO2 reconstruction from boron isotopes and I think we could exploit ESMs to understand/quantify the DpCO2 aspect of the calculation. Doing this kind of exercise with a larger ensemble would be really good way to test this central assumption in surface ocean carbonate system based pCO2 reconstructions in the future.

For the present study, I'd be happy to provide some version of the figures below (perhaps in DpCO2 space, rather than pH) to include in the manuscript (say below Figure 1) if the authors thought it would be helpful and wished to redraw them and include them.

[Figure]

*Figure above (from Gray and Evans 2019) pH difference minus the mean ocean pH difference in the IPSL model between PI and LGM forcings. Most of the surface ocean shows very little pH change beyond the impact of changing atmospheric CO2, which is reassuring for trying to reconstruct atmospheric CO2.*

For the second carbonate system parameter the authors use ALK, taking a flat 175 umol/kg range about the modern value – how is this range distributed around modern value? Is it weighted more heavily to higher values to account for likely higher glacial ALK a la Martinez-Boti 2015? They suggest the ALK variations needed would be too large to be sole cause of discrepancy, but we I guess cannot rule out more minor/systematic ALK changes over G-IG cycle cannot explain some part of discrepancy.

Detailed comments
line 58 – dissolved inorganic carbon (DIC)
line 138 – I think you could sure up this assumption or try to quantify the likely uncertainty in DpCO2 using ESM model output (see point above)
line 153 – Rebotim et al 2017 (biogeosciences) is a good reference for G. ruber habitat depth

line 198 – is 0.25% the relative concentration of the TPB? It could still have a big effect if it has a funky d11B, so better to report the absolute per mil values of correction
line 201 – with sacc or without sacc?
line 237 – I think using Anand needs some real justification (see discussion/figure above)
line 241 – an aside as you are not accounting for dissolution downcore, but it should really be parameterized as a function of Omega rather than depth
line 246 – I think more details are need of the method you used for the pH corrected approach, and greater discussion of the results are warranted later
line 254 – how is the ALK range distributed around modern value
line 273 – why not use the ESM output to try to quantify the likely DpCO2 uncertainty?
line 390 – sacc or no sacc?
line 431 – im not sure this is the reason Pacific CaCO3 preservation increases during glacials, as there is a lot of evidence for reduced ventilation of the deep Pacific in glacials (e.g. Anderson et al 2019). An ALK increase due to the reduction of CaCO3 burial in the deep Atlantic seems more likely (e.g. Cartapanis et al 2018), but a lot of this discussion seems somewhat superfluous. Its enough to say that the fragmentation should give an indication of relative changes in dissolution and describe the patterns seen.
line 461 – why?
line 467 – in Gray et al 2018 we found no systematic Mg/Ca offset between morphotypes across the Indian and Atlantic
line 483 – I really think ESMs can be useful here
line 524 – w/ or w/o sacc?
line 545 – really no TE data?
line 595 – worth noting optimized calibration is still very similar to other planktic calibrations i.e. it doesn't require something radically different to what we might expect
line 623 – the disequilibrium pump has garnered a lot of attention recently and is worth noting (e.g. Egglestone and Galbraith, 2018)

figure 1 – better to use red-white-blue colour scheme for positive and negative anomalies. why not also add a map of modelled LGM-PI DpCO2 differences below with the core sites indicated?
Figure s14 – why not plot on the other planktic lines to enable comparison with optimized line

---

## Author Response (AR1)

We respond to the reviewers in green and write in blue the text added to the manuscript refering to lines from the tracked change revised manuscript.

Review #1

de la Vega and co-authors present two beautiful new records on paleo-CO2 reconstructions that support the standing of the boron isotope proxy as a reliable indicator for paleo-CO2. The authors assess several aspects of the proxy, including dissolution and proxy calibration. The findings from these assessments are not entirely new, but that does not diminish the value of the manuscript. I have several recommendations for the authors to further improve the manuscript:

We thank the reviewer for their feedback and respond to each comment bellow.

The $Dlog_{10}$ $CO_2/DpH$ approach to estimate paleo-CO2 is an interesting one but the authors already indicate that it is only useful for short time scales, similar to the original study of Hain et al. 2018. This is somewhat disappointing for deep-time studies, where we really do not have a good sense of a second parameter of the carbon system. It would be helpful if the authors could discuss how this approach adds to data that we can already assess from ice cores over the past 800 kyr.

The advantage of this approach is that we compare $\delta^{11}B$-derived pH and CO2 forcing (derived from ice core CO2 od the same age) and to show the near-linear relationship between the two variables. Beyond 800 ky, in the absence of ice core CO2 data, we can use the reconstructed pH from boron isotopes and apply this linear relationship to reconstruct CO2 forcing.

We note that here only pH is needed and not the full calculation of $\delta^{11}B$-derived CO2 that requires the assumption on a second carbonate parameter. As shown in equation (5), the calculated pH is dependent on $\delta^{11}B_{borate}$, $\alpha_B$ (both known) and $\delta^{11}B_{sw}$, that has to be reconstructed beyond the last 2-3 million years due to the residence time of boron in the ocean (10-20 Myrs). However, as detailed in Hain et al (2018), the uncertainty associated with $\delta^{11}B_{sw}$ has a minor effect on delta $\Delta pH$ (see for example their Figure 4).

Whilst the requirement for a short time scales (i.e. <400 kyr) is still needed (to remove long-term variations in $\Delta DIC$ and $\Delta logHCO_3^-$), this doesn't prevent the reconstruction of $\Delta FCO_2$ for much of the geological record, even in the face of uncertainty in $\delta^{11}Bsw$.

The data added here to the compilation of Hain et al (2018), allows for a richer dataset that allows to compare pH to ice core CO2 and compare the updated linear relationship with the basic formalism $\Delta F/\Delta pH = -12.3 \ W/m^2$ and CaCO3, DIC, and Temperature driven relationships.

If the $\Delta FCO_2 /pH$ relationship was constant for the last 800 kyrs we have no reason to believe it should change across other orbital cycles (e.g., during the Mid Pleistocene Transition). The increase in higher resolution $\delta^{11}B$ record over the Plio-Pleistocene will allow to evaluate CO2 climate forcing with different climate background.

We have added in the introduction (line 11-117) the following text to clarify the point about slope comparison:

"Furthermore, given the principal drivers of the glacial-interglacial CO2 cycles (e.g. change in water mass, sea-ice cover, the soft tissue pump, the solubility pump, the CaCO3 counter pump and the disequilibrium pump; see Sigman et al., 2010; Hain et al., 2010, 2014 for a full review), will impact the $\Delta log_{10}CO_2/\Delta pH$ relationship in different ways, comparing the slope of the regressed $\Delta F/\Delta pH$ line from data to theoretical endmembers (temperature, DIC, CaCO3) could allow the primary controlling mechanisms during Glacial-Interglacial (G-IG) cycles to be deciphered."

Site selection: The authors keep using the same sediment records that they have been using for many years but Figure 1 demonstrates that neither site is ideal. This raises questions about the utility of using either site to "calibrate" the proxy and the reconstructed CO2 shown in Fig. 3 shows more extreme deviations from glacial and

interglacial CO2 extremes in the ice cores than site 999. An ideal site would be located in the vast ocean areas that are shaded green in Fig. 1. The final reconstructions are still impressive, but require corrections for a CO2 disequilibrium of which we cannot be certain that it remained constant through time. This caveat should be considered throughout the manuscript, as a source of uncertainty that affects all aspects of the study, including the calibration.

We acknowledge that we have focused much of our work at ODP 999, but this is the first study to produce $\delta^{11}B$ data for ODP 871. The reason we chose these sites is that they have sufficient *G. ruber senso stricto* for our purposes (1-2 mg), they have a priori determined age models based on $\delta^{18}O$ , have sufficiently high sedimentation rates to allow for the high-resolution sampling we carry out here, they are relatively shallow (<2800m) and preservation (although explored in detail in our manuscript) has been previously described as good or better. We agree with the reviewer that a site in the ocean gyres would be ideal in terms of $CO_2$ disequilibria but it is important to note that such sites would be less ideal in other ways. Most importantly for this study, the sites located beneath the gyres tend to either be outside of the range of *G. ruber* ss geographical distribution and/or have very low sedimentation rates. Furthermore, the gyres tend to be situated above the deepest parts of the ocean such that the carbonate sediments below them are typically below the regional lysocline/CCD. Sites ODP 999 and ODP 871 are therefore a necessary compromise with respect to their disequilibria. Importantly both sites are far from upwelling locations today, and are oligotrophic with $CO_2$ disequilibria less than 20 ppm. Our assumption of constant levels of disequilibria (as is commonly done) is, however of fundamental importance as the reviewer notes. As a result, we discussed this specific point in section 4.4. where we provide recommendations for future studies. We have also attempted to address this point directly in section 4.2.2 by using records of surface $\delta^{13}C$ and $\delta^{18}O$ in *G. ruber*. These two variables should be impacted if upwelling is happening at either core sites (lighter $\delta^{13}C$ and heavier $\delta^{18}O$, for colder nutrient rich waters) . In the absence of any visible deviation from the expected change during G-IG variations, we infer that upwelling had a minimal impact on our sites over the study interval. We however acknowledge that this interpretation could be further constrained by having records of productivity at each site (e.g., opal fluxes, alkenone concentration, Ba excess). In the absence of such records, we infer our conclusions based on planktonic $\delta^{13}C$ and $\delta^{18}O$ only. Also see reply below (about SST in Figure S9) about upwelling at site 871.

In order to better reflect the uncertainty in disequilibrium, in the revised manuscript we will fully propagate the uncertainty in this term using a conservative +/- 10 ppm (1SD) uncertainty on the disequilibrium correction. We will also refer to model outputs (Gray and Evans, 2019, suggested by reviewer 2) that shows that relative delta-pH does not change very much between the LGM and the pre-industrial at these two core sites.

We have added the following text (line 149-159):

Whilst we recognised that both sites have a minor disequilibrium, this is often a necessary compromise as areas of the ocean that are in strict equilibrium with the atmosphere are often located in the middle of oceanic gyres and tend to have deep sediments located under the lysocline, have a low sedimentation rate and/or are outside the preferred geographic habitat of *G. ruber*. Furthermore, we present surface $\delta^{18}O$ and $\delta^{13}C$ (site 871) and temperature (both sites) from *G. ruber* that provide insight into the potential influence of upwelling (see section 4.2.2) at these locations. Recent Earth System Model (IPSL-CM5A-MR) outputs (Gray and Evans, 2019) also show that relative pH difference at our core sites between the last glacial maximum (LGM) and the pre-industrial (PI), compared to the ocean average pH difference are close to 0, giving confidence that changes in local disequilibrium are unlikely to drive large changes in our $CO_2$ reconstructions (at least during the last glacial period).

Stable isotope record: The authors used only 10 planktic foraminifer shells for each sample in this record, which is a small number given the geochemical variability from shell to shell and bioturbation. Even laser ablation studies in laboratory culture, where specimens experience well controlled, constant environmental conditions, use at least 12-25 shells to overcome interspecimen variability (see e.g., Holland et al. 2020). It would have been better if the authors had picked larger samples for boron isotope analyses, crushed and homogenized them and then taken a small split for stable isotope analyses. While it would be asked too much to replicate the record with a larger shell number per sample at this time, the authors should mention that this sample size is not ideal, so that other researchers do not use it as a guideline. This also reflects on the genotype comparison (Fig. S5), which might have shown more significant results with a larger, more suitable sample size.

We appreciate the reviewer concerns on this point. It is normal procedure in our lab to measure a mixture from the ~200 or so foraminifera specimens to measure for oxygen isotopes alongside the boron isotopes". However in this case (where we ran $\delta^{18}O$ after $\delta^{11}B$) we decided to concentrate on precise and well defined morpho-types and thus limited the samples to 10 individuals of each. We recognise that this will lead to more variability in the record, but this is exactly what we are looking for to identify diagenetic alteration or preservation bias, hence the noisier record may actually be more informative in this instance. This will now be made clear in the text.

We have added the following text (line 189-192):

While this number of specimens is lower than classically done for $\delta^{18}O$ and $\delta^{13}C$ analysis, it provides power for the identification of species-specific preferential diagenetic alteration, which may have occurred in the sediment and it was sometimes necessary due to the scarcity of some of the *G. ruber* spp morphotypes.

Age models: The authors generate new data and display them in Fig. 2E,F but do not really discuss them. The figure caption describes a species correction but it is not clear how that correction has been determined and if it has already been applied to the displayed data or was applied afterwards. LR04 provides no guideline on this, as far as I can tell. Figure 2 shows site 999 data are too low compared to LR04 but site 871 data fall on LR04. This means that at least one of these records deviates from LR04, and the cause for this deviation (and the choice of species offset) should be discussed.

Given the benthic foraminiferal isotope records are only used for age correlation purposes (to provide a constraint independent from planktic $\delta^{18}O$ and $\delta^{11}B$ records) the precise offset used is unimportant, as long as it is consistent. Of course we recognise the importance of stating the correction applied so that others can go on to reuse and reinterpret the data. A correction of +0.47 was applied to *Cibicidoides wuellestorfi* at site 999 following Marchitto et al. (2014). The ODP 871 samples are measured on *Uvigerina peregrina* and thus required no correction. This is now clearly stated in the figure caption and the supplementary tables and we thank the reviewer for bring this important point forward.

We've added/modified the following text (line 198-205):

At Site 999, the age was determined by developing a new *Cibicidoides wuellerstorfi* benthic $\delta^{18}O$ record. The initial age model at Site 871 was refined by measuring $\delta^{18}O$ on the benthic species *Uvigerina peregrina* (50 µg of 3-5 mixed, crushed and homogenised specimens) measured on a Thermo KIEL IV Carbonate device at the University of Southampton, Waterfront Campus. These new $\delta^{18}O$ data (Figure 2) were then tuned to the benthic $\delta^{18}O$ LR04 stack (Lisiecki and Raymo, 2005) using Analyseries (Paillard et al., 1996). A correction of +0.47 was applied to the $\delta^{18}O$ *Cibicidoides wuellestorfi* at ODP Site 999 following Marchitto et al. (2014).

Dissolution experiment: This is the weakest part of the entire manuscript. The authors do not describe which sediment samples they used for the experiment. What makes those samples ideal for such an experiment? The lack of shell weight data is detrimental and essentially prevents any confidence in the data that have been collected. How large was the volume of acidified fluid? Is it possible that it got saturated right away, was there any dissolution at all? Here again the authors should say more clearly that their experiment falls short on several fronts. They do, but still interpret the data, which does not seem justified. Dissolution in acid and deionized water is likely very different from dissolution in corrosive seawater, so there really is little value in the experiment and associated data. The discussion draws mostly from earlier, much better dissolution studies, which serve the authors' purpose well, so the dissolution experiment could just as well be removed from this study without any impact on the discussion or results. The concern about including such an experiment is that it may lead others to follow the example, which would be unfortunate.

The samples used are from two intervals of ODP site 871 (age 182 ky and 160.81 ky), and were chosen because of the high abundance of forams at that site and intervals that allowed multiple repeats of the same material of the same age. We very regretfully could not measure elemental data for *T. sacculifer* due to machine down time for a significant period of time, after samples were dissolved. However weight and elemental data were obtained for the repeat leaching experiment on *G. ruber*. No change in weight and elemental composition were observed after 10 hours of leaching (in 1ml of acidified fluid, like all other leaching experiments) for *G. ruber*. Whilst this shows that indeed, no dissolution has occurred for this species, it shows some relative robustness to dissolution compared

to *T. sacculifer* that showed a significant δ¹¹B fractionation after only 6 hours of leaching. We however acknowledge weight and trace element data for *T. sacculifer* is a crucially needed addition. In the absence of these data, we agree with the reviewer and will remove this auxiliary data until further constraints can be obtained.

Text has been rearranged line 646 to 678.

Temperature estimates: The authors use both pH-corrected and uncorrected calibrations to translate Mg/Ca to SST. In the end they use the uncorrected estimates for calculating CO2 but it is not discussed why this should be the better choice. This is particularly striking after multiple studies have highlighted the pH dependence of Mg/Ca in *G. ruber*. The authors should discuss why they think this is the better approach following line 461. This choice affects the downcore calibration for d11B and deserves more attention.

We thank the reviewer for their comment. This point was also raised by reviewer 2. We have conducted sensitivity tests with seven treatments of Mg/Ca or temperature: (1,2) Gray et al., 2018 with and without a pH correction, (3,4) Gray et al., 2018 using Mg/Ca corrected for depth-dependent dissolution, with and without a pH correction, (5,6) Anand et al., (2003) with and without a depth correction, and (7) constant temperature. The results are displayed bellow (Figure R1).

[Figure]

Figure R1. Effect of various temperature treatments (top), on δ¹¹B-derived CO₂ (middle) and CO₂ offset to the ice cores (bottom). Left: site 999, right: site 871. Temperature treatments are: G18 (Gray et al., 2018 no correction), G18 pH corr (Gray et al., 2018 with pH correction), G18 depth corr (Gray et al., 2018 with Mg/Ca corrected depth), G18 pH-depth corr (Gray et al., 2018 with depth and pH correction), Anand (Anand et al., 2003, no correction), Anand depth corr (Anand et al., 2003, Mg/Ca corrected for depth), Temp cst (constant temperature of 26°C).

We have chosen the Mg/Ca treatments that accounts for pH effect on Mg/Ca and yields the closest agreement between coretop and modern temperature from Glodap v2 at both sites (Note coretop at 871 is not displayed and the most recent Mg/Ca from Dyez and Ravelo, 2013 was used). This treatment is with a pH correction (Gray et al., 2018) and Mg/Ca corrected for depth/dissolution.

Following suggestions from both reviewers we will update all data in the revised manuscript including $\delta^{11}$B-derived $CO_2$ to reflect this change in Mg/Ca treatment. The resulting average $CO_2$ offsets for each Mg/Ca treatment are displayed below (Table R1).

| core | Mg/Ca treatment | DpCO2 | 2sd | core | Mg/Ca treatment | DpCO2 | 2sd | Average DpCO2 for both records |
|------|-----------------|-------|-----|------|-----------------|-------|-----|-------------------------------|
| 999 | T Gray18 | -27.03 | 40 | 871 | T Gray18 | -28.21 | 52 | -27.62 |
| 999 | T Gray18 pH corr | -3.87 | 39 | 871 | T Gray18 pH corr | -6.40 | 47 | -5.14 |
| 999 | T Gray18 depth corr | -6.77 | 43 | 871 | T Gray18 depth corr | -3.30 | 54 | -5.03 |
| 999 | T Gray18 pH corr, depth corr | 12.06 | 41 | 871 | T Gray18 pH corr, depth corr | 13.89 | 51 | 12.98 |
| 999 | T Anand03 | 6.64 | 45 | 871 | T Anand03 | 1.37 | 54 | 4.00 |
| 999 | T Anand03 depth corr | 23.28 | 48 | 871 | T Anand03 depth corr | 20.88 | 57 | 22.08 |
| 999 | T constant | 6.33 | 46 | 871 | T constant | 0.23 | 53 | 3.28 |

Table R1. Effect of various Mg/Ca-derived temperature calibrations on CO2 offset (DpCO2). The line in green is the chosen updated calibration.

We've added the following text (line 275-304), updated Figures 2, 3, 4, 5 and 6 and added in the supplements Table S1, S2 and Figure S1.

The sea surface temperature (SST) values necessary to calculate pK$_B$ in equation (5) were determined at both sites using the Mg/Ca of *G. ruber* (Dyez and Ravelo, 2013) including a depth-dependent dissolution correction for each site (following Dyez and Ravelo, 2013 for Site 871 and Schmidt et al., 2006 for Site 999) and a pH correction using the iterative approach of Gray and Evans (2019) to account for the observed pH effect on Mg/Ca in *G. ruber* producing higher apparent sensitivity of Mg/Ca during glacial cycles (Gray et al., 2018).

Mg/Ca was corrected for depth-dependent dissolution at Site 871 using the following equation (Dyez and Ravelo, 2013):

$$\frac{\text{Mg}}{\text{Ca}}(corrected) = \frac{\text{Mg}}{\text{Ca}}(measured) + 0.26 * depth + 0.52 \ (6)$$

Mg/Ca from Site 999 was corrected following Schmidt et al. (2006):

$$\frac{\text{Mg}}{\text{Ca}}(corrected) = \frac{\text{Mg}}{\text{Ca}}(measured) + 0.66$$

To evaluate the effect of various Mg/Ca treatment on temperature and calculated $CO_2$, we performed seven sensitivity tests (Table S1) with Mg/Ca-derived SST using the calibrations of: (1) Gray et al. (2018) temperature-dependent only (global calibration), (2) Gray and Evans (2019) with a pH correction; (3) Gray et al. (2018) temperature-dependent with Mg/Ca corrected for depth-dependent dissolution; (4) Gray and Evans (2019) with Mg/Ca corrected for depth-dependent dissolution and pH correction; (5,6) Anand et al. (2003) with and without a depth correction; and (6) with temperature kept constant (26°C).

The differences in SST and resulting $CO_2$ can be substantial (Figure S1, Table S2): up to 6 degrees and ~50 ppm, respectively, between the Gray et al. (2018) calibration uncorrected for pH and the Anand et al. (2003) calibration corrected for dissolution. We have chosen the Mg/Ca treatments that accounts for pH effect on Mg/Ca and yields the closest agreement between coretop at both sites and modern temperature from Glodap v2 (Lauvset et al., 2022) This treatment is with a pH correction and Mg/Ca corrected for depth dependent dissolution. Choosing this approach is justified considering (1) the strong offset between Anand et al. (2003) multi-species Mg/Ca-Temperature calibration and the more recent *G. ruber* compilation of Gray et al., (2018); (2) the effect of pH correction as shown in Gray et al., (2018) and Gray and Evans (2019); (3) the suggested influence of dissolution on Mg/Ca (Dyez and Ravelo, 2013; Schmidt et al., 2006) and (4) the better agreement between coretop and modern SST at each site when using a pH and depth correction (Figure S1).

CO2 offsets statistics updated in line 442-444 and throughout the text:

the mean offset from the ice core $CO_2$ for a combination of the two records is 13±46 (2$\sigma$) ppm showing that there is a minor overestimation of $CO_2$ using the boron method yet it agrees on average well within uncertainty. The RMSE of the $CO_2$ offset for the combined record is 26 ppm.

in Line 617-620:

Periods of high fragmentation at ODP Site 999 and 871 correspond to a positive $CO_2$ offset 65 and 75% of the time respectively, and 35 and 25% of the time to a negative or no (i.e. ±10 ppm) $CO_2$ offset, (note that values ±10 ppm were omitted in the criteria for positive or negative offset).

And line 736-738 with the optimised calibration:

Intervals of high fragments occur 5% and 33% of the time, at Sites 999 and 871, respectively, during positive $CO_2$ offsets (and 95 and 67% of the time during negative or no offset to the ice cores).

-Figure S9 also discusses "anomalous" temperature estimates but no discussion is provided as to what constitutes such an anomalous deviation. What is the point of reference? How does the SST record compare to sites of similar latitude but outside of potential upwelling areas? (e.g., gyre sites). How do we know that SST was not cooler than expected?

We thank the reviewer for this relevant comment. A comparison between site 871 SST with an Mg/Ca-derived SST record from the Western Pacific warm pool site MD97-2140 (de Garidel-Thoron et al., 2005, figure R2) a location outside of the upwelling from the Pacific cold tongue, shows that the periods of high $CO_2$ offset (230, 290, 380 ky) are not associated with relatively cold periods at ODP site 871, which suggests that they are not related to upwelling (since upwelled water should be cold and have high $CO_2$). We note that this comparison needs to be caveated by the different treatments of Mg/Ca in de Garidel-Thoron et al. (2005). Site 871 was also chosen as it is a previously studied site (Dyez and Ravelo, 2013, 2014). These studies compared temperature records of the on-equatorial ODP site 806 and the off-equatorial site 871, and concluded site 871 was unlikely to be impacted by upwelling due to its deep thermocline and the stronger cooling observed at 806 possibly linked to upwelled thermocline waters at this site forced by Northern hemisphere summer insolation. Similar to the reviewer's previous comments about disequilibrium, we also acknowledge that this interpretation could be further constrained by having records of productivity at each site (e.g., opal fluxes, alkenone concentration, Ba excess).

Upwelling at site 999 is thought to happen seasonally in the modern with associated $CO_2$ flux of +20 ppm (Olsen et al., 2004) that we correct for in downcore $CO_2$ reconstruction. Changes in upwelling at Site 999 may have occurred in the past in relation to the position of the ITCZ (see discussion in Foster, 2008). Foster and Sexton (2014) have also reconstructed $CO_2$ zonally across the equatorial Atlantic and the Caribbean and show that for the last 30 ky at least, Site 999 has remained in equilibrium with the atmosphere. Whilst SST is a first order constraint on upwelling, future studies need to focus on paired proxies of temperature and productivity to evaluate change in local $CO_2$ fluxes. As mention above in the reviewer's comment about disequilibrium. we will fully propagate the uncertainty in this term using a conservative +/- 10 ppm (1SD) uncertainty on the disequilibrium correction.

[Figure]

Figure R2. CO₂ offsets (or residual) and updated SST (pH and depth correction on Mg/Ca), and comparison with SST record from the Western Pacific warm pool site MD97-2140 (De Garidel-Thoron et al., 2005).

We've added the following text (lines 601-611), and added Figure S8 .

The Mg/Ca-derived SST record of nearby Site MD97-2140 (Figure S8) from the Western Pacific warm pool (de Garidel-Thoron et al., 2005) a location outside of the upwelling from the Pacific cold tongue, confirms this view in that the periods of high CO₂ offset at Site 871 are not associated with relatively cold periods at site MD97-2140. Equally, no SST anomaly was identified at ODP 999 to be coincident with the intervals of high residual CO₂ . Foster and Sexton (2014) have also reconstructed CO₂ zonally across the equatorial Atlantic and the Caribbean and showed that while enhanced disequilibrium was detected in the eastern Atlantic, for the last 30 ky at least, Site 999 has remained in equilibrium with the atmosphere. This suggests the CO₂ anomalies revealed in Figure 5 are not the result of enhanced local disequilibrium via sub-surface water mixing. Whilst SST is a first order constraint on upwelling, we acknowledge future constrains are needed using paired proxies of local CO₂, temperature and productivity to evaluate changes in local CO₂ fluxes.

-CO2 forcing: Figure 4 shows a nice correlation between DFCO2 and pH but the data deviate at least 5-6 times from the regression lines and their uncertainty, if that is displayed by the grey shading, it does not capture the true data variability. Based on the scatter around the lines, how many d11B data do the authors suggest are needed to provide a single reasonable estimate of DFCO2 for a given point in time? Fig. S7 suggests an uncertainty of +/- 0.3-0.8 W/m2, which is clearly an underestimate given the data scatter and requires an assessment of the number of data needed to provide such a minimal uncertainty.

The key variable to look at is the goodness of fit MSWD. The higher the uncertainty in the data, the better the MSWD is ( i.e. close to 1) since points furthest away are then more consistent with the regressed line.  As demonstrated in Figure S7 and updated below (Figure R3) with new temperature estimates, a lower uncertainty assigned to pH gives a poorer MSWD: the data is over dispersed, given their low uncertainty. This variable is more informative about the data dispersion than the envelope around the York regression. The number of $\delta^{11}$B data points helps to assess how accurate the basic formalism  ($\Delta F/\Delta pH$ =-12.3 W/m² ) is. Providing a minimum number of points to estimate $\Delta FCO_2$ would be rather arbitrary. With the current data set, the average $\Delta FCO_2$

deviation from the best fit line is -0.13 W/m². The main idea of the approach is (1) to validate the basic formalism and (2) to evaluate where the regressed slope derived from $\delta^{11}$B-pH data falls relative to the temperature, $CaCO_3$ and DIC driven slopes. The richer the dataset the more accurate the regressed slope is. Improvement in high resolution records as well as analytical uncertainties of $\delta^{11}$B (which accounts for the main uncertainty in pH), will refine the accuracy of the data-derived $\Delta$F-$\Delta$pH relationship and how much it deviates from the basic formalism. The original study of Hain et al (2018) reconstructed $\Delta FCO_2$ with an uncertainty empirically determined by the $\Delta FCO_2$ range between the lowest and highest endmember slope of the $\Delta$pH-to-$\Delta$F relationship (i.e. $CaCO_3$ and DIC change), this accounts for both uncertainty in $\delta^{11}$B-derived pH as well as the conversion from $\Delta$pH to $\Delta$F. With the benefit of more data and a more thorough consideration of the pH reconstruction uncertainty we empirically determine the pH/$\log CO_2$ relationship to be closer to the steep DIC endmember. We think it is a conservative approach to consider the full endmember range, which allows to confidently reconstruct $\Delta$F in the past, but our data suggests this range can be reduced with further work.

[Figure]

pH uncertainty from $\delta^{11}$B borate only (1$\sigma$):
$\Delta F/\Delta pH$= - 13.7 ± 0.3 W/m2
mswd=5.34

pH uncertainty from Monte Carlo simulation (1$\sigma$):
$\Delta F/\Delta pH$= - 17.2 ± 1 W/m2
mswd=0.85

Figure R3 (Figure S7 in revised manuscript). Updated $\Delta FCO_2$-$\Delta$pH relationship when using Mg/Ca pH and depth corrected.

-Downcore calibration: This is an interesting approach that could be applied to species that have not yet been calibrated in culture, and it is an approach that could rival coretop calibrations because the modern pH range in surface seawaters is generally too small to allow for a high-quality calibration. However, here again I wonder how many cores and data should be included in a calibration exercise, and whether the downcore calibration is really stronger than the existing culture calibration for *G. ruber*. To do so, I would recommend that the authors generate a new calibration for each core site and then apply that calibration to the respective other core site. How different are the calibrations from each other and from the culture calibration, and do both calibrations improve the match of the CO2 estimates to the ice cores?

We thank the reviewer for their comment and while we agree that this is a useful approach we did not want to give the impression this is the best way to deal with $\delta^{11}$B data going forward. As section 4.3 demonstrates there is considerable power in having an independent record of ocean pH and boron is unique among the $CO_2$ proxies in that it is not tied to the ice core $CO_2$ records in any way. We agree that for species without a culture or coretop calibration this approach would be useful, we have used the combined record here principally for the sake of illustration.

However, since the two records have $CO_2$ offset that occur at similar periods (Figure S3), it is unlikely that a downcore calibration conducted on each site greatly differs from the combined records. To showcase this, we include here the same exercise performed on each separate record and display all the slopes and derived $CO_2$ offset (figure R4 and table R2 below). These results show that each separate calibration do not show significant deviation in calculated $CO_2$ (the average $CO_2$ offset for 871 and 999 calculated with their respective downcore calibration is -5 ppm) from the calibration obtained from the combined two record (+4 ppm average offset).

We agree that this approach is dependent on a given record, since cores from different locations can differ from one another by having different oceanographic setting or dissolution history, and we discuss this in section 4.4.

[Figure]

| Downcore calibration | Slope | Intercept | Average CO2 offset (ppm) |
|---|---|---|---|
| All data | 0.713 | 6.492 | 4 |
| 999 only | 0.724 | 6.326 | -7 |
| 871 only | 0.710 | 6.532 | -3 |

Figure R4. $\delta^{11}B$-derived $CO_2$ and resulting $CO_2$ offset calculated using optimised calibration using combined data set from both cores and each separate data set from ODP 999 and 871.

Table R2. Slope, intercept and average CO2 offset for optimised calibration using combined data set from both cores and each separate data set from ODP 999 and 871.

We've added the following text (line 404-407):

To assess the effect of $\delta^{11}B$ records from different sites we performed this exercise using the combined records (from both sites 999 and 871), 999 only and 871 only (Figure S3 ) and show that using a record from one particular site or the combination of sites yields similar $CO_2$ offsets (Table S3) and so here we use the results from the combined sites.

-Finally, the authors should check spelling and grammar throughout, including names of authors whose work they cite. There are several typos throughout the manuscript, and in some cases incomplete sentences. Please check spelling in lines 42, 45, 74/75, 149, 150, 186, 215, 413, 433, 526, 570, 691, 693. The sentences in line 607-610 should be rephrased entirely.

This typos have been updated and the lines 607-610 (736-738 in revised manuscript) rephrased as follow (including updated numbers):

Intervals of high fragments occur 5% and 33% of the time, at Sites 999 and 871, respectively, during positive $CO_2$ offsets (and 95 and 67% of the time during negative or no offset to the ice cores).

-In summary, this study adds valuable confirmation to an already strong proxy. There is still room for improvement in this manuscript, mostly by clarifying certain choices, but also by assessing the paleo-calibration from different angles.

We thank the reviewer for their thorough constructive feedback.

Review #2

**Orbital CO2 reconstruction using boron isotopes during the late Pleistocene, an assessment of accuracy – de la Vega et al**

review by William Gray (william.gray@lsce.ipsl.fr)

Boron isotopes in planktic foraminifera are a widely used proxy for reconstructing ancient CO2. As other studies have done previously, de la Vega et al use ice core CO2 as a test of accuracy; to this end, they present new G-IG boron isotope data, including data from a new sediment core site in a location close to equilibrium with the atmosphere today. This is only the second record of its kind (i.e. a record from a site close to equilibrium and over multiple glacial cycles) and is a very welcome addition as it helps overcome the assumption that a single site (i.e. ODP 999) has remained in equilibrium with the atmosphere. They go on to assess the accuracy with which CO2 can be reconstructed, and possible causes of the (albeit relatively minor) discrepancies (dissolution, second carb system parameter, calcite-borate calibration). Overall, I found the manuscript to be clear, well structured, and well-reasoned. We thank the reviewer for his constructive positive feedback and respond to each comment below.

I have the following suggestions:

I think a greater exploration/discussion of the thermal influence on the carbonate/borate system (via the dissociation constants) in paleo CO2 reconstruction is warranted – basically, how sensitive is CO2 to accurate SST reconstruction? More sensitivity tests could be implemented - including keeping temperature constant throughout. Overall, I think further exploration/discussion of the thermal effects are warranted as, my guess is, this could be an important source of bias/uncertainty and it's helpful to understand what we need to improve. The authors mention they also use our iterative pH correction, and briefly mention this in the text, but I think it warrants further discussion; there is very good evidence from culture studies that pH influences Mg/Ca, and this is an influence we know is going to covary with atmospheric CO2.

The authors mainly rely on the calibration of Anand et al 2003 to derive Mg/Ca SSTs to use in their CO2 reconstruction. Although widely applied I don't think the calibration of Anand et al accurately describes the relationship between Mg/Ca and temperature (see figures 5 and 6 in Gray et al 2018). This is very apparent if you compare the measured CTD temperatures at the sediment trap site used by Anand, and the temperature calculated using Anand's own Mg/Ca data and their calibration line (see figure below). Using the calibration of Anand et the Mg/Ca are almost always too warm, and the seasonal cycle is about half of what it should be as winter temperatures are

4 degrees too warm (there is no way to explain this by 'sampling issues' as the Mg/Ca SSTs are warmer than any individual CTD measurement ever taken at the site in winter, and this is BATS so it's about the most heavily sampled place in the ocean). Basically, if you use Anand et al on their own data, you get the wrong answer. If the authors want to use Anand et al, I think there needs to be more justification (and 'it gives a better fit to the CO2 data/the core top SSTs' isn't a great reason).

[Figure]

*Figure above shows a comparison of Mg/Ca and CTD temperatures at the Sargasso Sea sediment trap site (the site used by Anand to derive their Mg/Ca calibration). I'm showing a LOESS fit to the Mg/Ca data (colored lines), rather than the individual data points to make it legible. The grey dots show all the individual CTD measurements ever taken at this site. The black line/grey shaded area is the expected temperature (and 95% CI) at the habitat depth of G. ruber.*

We thank the reviewer for this very important point drawing our attention to an alternative Mg/Ca calibration. This point is also mentioned by reviewer 1.

We have now conducted sensitivity tests with seven treatments of Mg/Ca or temperature: (1,2) Gray et al., (2018) with and without a pH correction, (3,4) Gray et al. (2018) using Mg/Ca corrected for depth-dependent dissolution,- with and without a pH correction, (5,6) Anand et al. (2003) with and without a depth correction, and (7) constant temperature. The results are displayed in the Figure below (Figure R1)

The differences in SST and resulting $CO_2$ can be substantial: up to 6 degrees and ~50 ppm, respectively, between the Gray18 calibration uncorrected for pH and the Anand et al (2003) calibration corrected for dissolution. We have chosen the Mg/Ca treatments that accounts for pH effect on Mg/Ca and yields the closest agreement between coretop at both sites and modern temperature from Glodap v2 (Note coretop at 871 is not displayed and the most recent Mg/Ca from Dyez and Ravelo, 2013 was used). This treatments is with a pH correction and Mg/Ca corrected for depth/dissolution.

Choosing this approach is justified considering (1) the strong offset between Anand's multi-species Mg/Ca-T calibration and the more recent *G. ruber* compilation of Gray et al., (2018), as the reviewer notes; (2) the effect of pH correction as shown in Gray et al., (2018) and Gray and Evans (2019); (3) the suggested influence of dissolution on Mg/Ca (Dyez and Ravelo, 2013; Schmidt et al., 2006) and (4) the better agreement between coretop and modern SST at each site when using a pH and depth correction.

A full discussion on the various treatments of Mg/Ca is not the scope of this study but in the revised manuscript we will describe and present the different Mg/Ca calculation and resulting $CO_2$, and justify our chosen approach. We will note that the choice of Mg/Ca calculation is an added cause of potential $CO_2$ offsets and the way SST is calculated varies amongst boron isotope studies. However, the updated SST here considering the effect of pH and dissolution on Mg/Ca does not affect the main conclusions of the paper and the $\delta^{11}B$ downcore calibration still allows improvement to the fit to ice core $CO_2$.

Figures and data have been updated to account for the new Mg/Ca treatment and the estimated $CO_2$ are now slightly higher than previously calculated with an average updated combined offset to the ice core $CO_2$ of 13 +/- 46 ppm .

The comparison between fragmentation index and the updated $CO_2$ (pH and depth corrected) in Table R1 shows that periods of high fragments and high $CO_2$ offset coincide 65 and 75% of the time at core 999 and 871, respectively with the calibration of Henehan et al. (2013), and 5 and 33% of the time with the optimised calibration and pH-corrected Mg/Ca-derived SST. Note that almost all $CO_2$ offsets are within error (95 % confidence envelope) of the 0 residual line and that between 20 (ODP 999) and 55 % (ODP 871) of the $CO_2$ offsets are within 10 ppm (Table R1). $CO_2$ offsets below 10 ppm are effectively zero and not included in the criteria of positive or negative $CO_2$ offset. This evaluation should therefore be treated with caution.

|  | 999 (calib H13) | | 871 (calib H13) | | | 999 (optimised calib) | | 871 (optimised calib) | |
|---|---|---|---|---|---|---|---|---|---|
| CO2 points associated with high FRAGMENTS | Nb of points | % occurrence | Nb of points | % occurrence | | Nb of points | % occurrence | Nb of points | % occurrence |
| negative CO2 offset | 3 | 15 | 0 | 0 | | 8 | 40 | 3 | 25 |
| positive CO2 offset | 13 | 65 | 9 | 75 | | 1 | 5 | 4 | 33 |
| within +/- 10 ppm | 4 | 20 | 3 | 25 | | 11 | 55 | 5 | 42 |
| Total | 20 | | 12 | | | 20 | | 12 | |

Table R1. Effect of various $\delta^{11}B$ calibration on the direction (positive or negative) of the $CO_2$ offset to the ice cores.

[Figure]

Figure R1. Effect of various temperature treatments (top), on $\delta^{11}B$-derived $CO_2$ (middle) and $CO_2$ offset to the ice cores (bottom). Left: site 999, right: site 871. Temperature treatments are: G18 (Gray et al., 2018 no correction), G18 pH corr (Gray et al., 2018 with pH correction), G18 depth corr (Gray et al., 2018 with Mg/Ca corrected depth), G18 pH-depth corr (Gray et al., 2018 with depth and pH correction), Anand (Anand et al., 2003, no correction), Anand depth corr (Anand et al., 2003, Mg/Ca corrected for depth), Temp cst (constant temperature of 26°C).

| core | Mg/Ca treatment | DpCO2 | 2sd | | core | Mg/Ca treatment | DpCO2 | 2sd | | Average DpCO2 for both records |
|---|---|---|---|---|---|---|---|---|---|---|
| 999 | T Gray18 | -27.03 | 40 | | 871 | T Gray18 | -28.21 | 52 | | -27.62 |
| 999 | T Gray18 pH corr | -3.87 | 39 | | 871 | T Gray18 pH corr | -6.40 | 47 | | -5.14 |
| 999 | T Gray18 depth corr | -6.77 | 43 | | 871 | T Gray18 depth corr | -3.30 | 54 | | -5.03 |
| 999 | T Gray18 pH corr, depth corr | 12.06 | 41 | | 871 | T Gray18 pH corr, depth corr | 13.89 | 51 | | 12.98 |
| 999 | T Anand03 | 6.64 | 45 | | 871 | T Anand03 | 1.37 | 54 | | 4.00 |
| 999 | T Anand03 depth corr | 23.28 | 48 | | 871 | T Anand03 depth corr | 20.88 | 57 | | 22.08 |
| 999 | T constant | 6.33 | 46 | | 871 | T constant | 0.23 | 53 | | 3.28 |

Table R2. Effect of various Mg/Ca-derived temperature calibrations on $CO_2$ offset (DpCO2). The line highlighted in green is the chosen updated calibration.

We've added the following text (line 275-304), updated Figures 2, 3, 4, 5 and 6 and added in the supplements Table S1, S2 and Figure S1.

The sea surface temperature (SST) values necessary to calculate $pK_B$ in equation (5) were determined at both sites using the Mg/Ca of *G. ruber* (Dyez and Ravelo, 2013) including a depth-dependent dissolution correction for each site (following Dyez and Ravelo, 2013 for Site 871 and Schmidt et al., 2006 for Site 999) and a pH correction using the iterative approach of Gray and Evans (2019) to account for the observed pH effect on Mg/Ca in *G. ruber* producing higher apparent sensitivity of Mg/Ca during glacial cycles (Gray et al., 2018).

Mg/Ca was corrected for depth-dependent dissolution at Site 871 using the following equation (Dyez and Ravelo, 2013):

$$\frac{Mg}{Ca}(corrected) = \frac{Mg}{Ca}(measured) + 0.26 * depth + 0.52 \ (6)$$

Mg/Ca from Site 999 was corrected following Schmidt et al. (2006):

$$\frac{Mg}{Ca}(corrected) = \frac{Mg}{Ca}(measured) + 0.66$$

To evaluate the effect of various Mg/Ca treatment on temperature and calculated $CO_2$, we performed seven sensitivity tests (Table S1) with Mg/Ca-derived SST using the calibrations of: (1) Gray et al. (2018) temperature-dependent only (global calibration), (2) Gray and Evans (2019) with a pH correction; (3) Gray et al. (2018) temperature-dependent with Mg/Ca corrected for depth-dependent dissolution; (4) Gray and Evans (2019) with Mg/Ca corrected for depth-dependent dissolution and pH correction; (5,6) Anand et al. (2003) with and without a depth correction; and (6) with temperature kept constant (26°C).

The differences in SST and resulting $CO_2$ can be substantial (Figure S1, Table S2): up to 6 degrees and ~50 ppm, respectively, between the Gray et al. (2018) calibration uncorrected for pH and the Anand et al. (2003) calibration corrected for dissolution. We have chosen the Mg/Ca treatments that accounts for pH effect on Mg/Ca and yields the closest agreement between coretop at both sites and modern temperature from Glodap v2 (Lauvset et al., 2022) This treatment is with a pH correction and Mg/Ca corrected for depth dependent dissolution. Choosing this approach is justified considering (1) the strong offset between Anand et al. (2003) multi-species Mg/Ca-Temperature calibration and the more recent *G. ruber* compilation of Gray et al., (2018); (2) the effect of pH correction as shown in Gray et al., (2018) and Gray and Evans (2019); (3) the suggested influence of dissolution on Mg/Ca (Dyez and Ravelo, 2013; Schmidt et al., 2006) and (4) the better agreement between coretop and modern SST at each site when using a pH and depth correction (Figure S1).

CO2 offsets statistics updated in line 442-444 and throughout the text:

the mean offset from the ice core $CO_2$ for a combination of the two records is 13±46 (2σ) ppm showing that there is a minor overestimation of $CO_2$ using the boron method yet it agrees on average well within uncertainty. The RMSE of the $CO_2$ offset for the combined record is 26 ppm.

in Line 617-620:

Periods of high fragmentation at ODP Site 999 and 871 correspond to a positive $CO_2$ offset 65 and 75% of the time respectively, and 35 and 25% of the time to a negative or no (i.e. ±10 ppm) $CO_2$ offset, (note that values ±10 ppm were omitted in the criteria for positive or negative offset).

And line 736-738 with the optimised calibration:

Intervals of high fragments occur 5% and 33% of the time, at Sites 999 and 871, respectively, during positive CO₂ offsets (and 95 and 67% of the time during negative or no offset to the ice cores).

-For the leaching experiment is *T. sacculifer* with or without sacc? This is really important as the lines of argument regarding the differential dissolution of the gametogenic calcite versus the rest of test only hold if it has a sacc (note there are similar arguments for a d18O dissolution effect in this species). The authors mention they didn't assess weight loss (a shame), but are there really no TE data for these samples? Was an aliquot not analysed as part of the boron isotope analysis to check for cleaning etc? If TE data are available it would be really informative to show how Mg changes with the leaching, as having some metric to be able compare to real world samples would make these leaching experiments much more useful.

*T. sacculifer* was picked with sacc-like final chamber but as mention in our response to reviewer 1, we very regretfully could not measure elemental data for *T. sacculifer* due to machine down time for a significant period of time after samples were dissolved. However, weight and elemental data were obtained for the repeat leaching experiment on *G. ruber*. No change in weight and elemental composition were observed after 10 hours of leaching for *G. ruber*. Whilst this shows that indeed, no dissolution has occurred for this species, it shows some relative robustness to dissolution compared to *T. sacculifer* that showed a significant $\delta^{11}B$ fractionation after 6 hours of leaching. We however acknowledge weight and trace element data for *T. sacculifer* would be a very welcomed addition. For this reason, we have decided to remove these results from the manuscript and we will aim to complement these data in the future.

-For the uncertainty and the discussion of changing DpCO2 its worth noting that Earth System Models under glacial forcings typically simulate very small changes in DpCO2 – this can be seen in the figures below which show the pH difference minus the mean ocean pH difference in the IPSL model between PI and LGM forcings (taken from Gray and Evans 2019) – basically most of the ocean just reflects the change in atmospheric CO2 (95% range within ±0.05 units, equivalent to about ±40 uatm DpCO2). There is almost no residual pH change at the two sites used in the present study. I think this is really encouraging for paleo CO2 reconstruction from boron isotopes and I think we could exploit ESMs to understand/quantify the DpCO2 aspect of the calculation. Doing this kind of exercise with a larger ensemble would be really good way to test this central assumption in surface ocean carbonate system based pCO2 reconstructions in the future.

For the present study, I'd be happy to provide some version of the figures below (perhaps in DpCO2 space, rather than pH) to include in the manuscript (say below Figure 1) if the authors thought it would be helpful and wished to redraw them and include them.

[Figure]

*Figure above (from Gray and Evans 2019) pH difference minus the mean ocean pH difference in the IPSL model between PI and LGM forcings. Most of the surface ocean shows very little pH change beyond the impact of changing atmospheric CO2, which is reassuring for trying to reconstruct atmospheric CO2.*

We thank the reviewer for this input. This is an encouraging result that adds confidence to the fundamental tenet that a given site has remained in equilibrium with the atmosphere in the past. We will refer to this published figure when introducing the modern ocean-atmosphere pCO₂ map (Figure 1).

Added text (line 154-159):

Recent Earth System Model (IPSL-CM5A-MR) outputs (Gray and Evans, 2019) also show that relative pH difference at our core sites between the last glacial maximum (LGM) and the pre-industrial (PI), compared to the ocean average pH difference are close to 0, giving confidence that changes in local disequilibrium are unlikely to drive large changes in our $CO_2$ reconstructions (at least during the last glacial period).

For the second carbonate system parameter the authors use ALK, taking a flat 175 µmol/kg range about the modern value – how is this range distributed around modern value? Is it weighted more heavily to higher values to account for likely higher glacial ALK à la Martinez- Boti 2015? They suggest the ALK variations needed would be too large to be sole cause of discrepancy, but we I guess cannot rule out more minor/systematic ALK changes over G-IG cycle cannot explain some part of discrepancy.

The uncertainty of 175 µmol/kg around alkalinity is equal on either side of the central value (now specified line 327). The figure R2 bellow shows the $CO_2$ output for TA uncertainty distributed uniformly (black) or weighted more heavily to the higher value (red). The results for both treatment are close and overall show a slightly higher $CO_2$ offset to the ice core in the case of asymmetrical uncertainty (+ 17 ppm offset on average for both cores, vs. +13 ppm for a uniform uncertainty). This is due to the higher number of simulation in the Monte Carlo that will lie above the central alkalinity value in case of a -25/+75 envelope, and this has the effect of increasing $CO_2$.

Whilst the reasoning on a higher uncertainty bond during glacials is sensible (based on model outputs, Toggweiler, 1999; Hain et al., 2010), we think it is more conservative to keep a symmetrical large uncertainty to allow for the uncertain factors in contributing to alkalinity change over G-IG changes (carbonate burial, carbonate pump, soft-tissue pump).

[Figure]

Figure R2. Effect of total alkalinity (TA) uncertainty on reconstructed $CO_2$, symmetrical +/- 175 µmol/kg (black line and grey envelope) and asymmetrical -25/+75 µmol/kg (dotted red line and pink envelope).

Detailed comments
line 58 – dissolved inorganic carbon (DIC). "Total" has been removed.

line 138 – I think you could sure up this assumption or try to quantify the likely uncertainty in DpCO2 using ESM

model output (see point above).| line 483 – I really think ESMs can be useful here. | line 273 – why not use the ESM output to try to quantify the likely DpCO2 uncertainty? We will refer to this model output that adds confidence to the assumption that a given site has remained in equilibrium with the atmosphere in the past.

line 153 – Rebotim et al 2017 (biogeosciences) is a good reference for G. ruber habitat depth. We thank the reviewer, this reference has been added.

line 198 – is 0.25% the relative concentration of the TPB? It could still have a big effect if it has a funky d11B, so better to report the absolute per mil values of correction. Yes, 0.25% is the relative concentration of the TPB given a typical TPB contribution of 50 pg and a sample size of 20 ng. We however use a long term median of TPBs measured over multiple years (-7 permille) rather than the $\delta^{11}B$ of TPB measured on the day of column since a single $\delta^{11}B$ measurement has poor confidence given the small amount of boron in a TPB samples ( 40-100 pg) and the analytical methods used here. We also update a mistake in the manuscript that omitted to mention TPB correction conducted on the new samples from ODP site 999, for which TPB values ranged 70 to 100 pg. This resulted in a TPB fraction of 0.3 to 2.3% of the samples and a $\delta^{11}B$ correction of 0.1 to 0.7 permille. This will be added to the text in the revised manuscript.

Text added line 229-234:

A total procedure blank (TPB) was conducted for each batch of samples and typically ranged from 0-100 pg. which represents a blank contribution of up to 2.3% (for samples containing ~10-20 ng of boron). Most samples had a TPB below 40 pg and were not corrected.  Two batches had a TPB of 70 and 100 pg for which we corrected using a long-term median TPB $\delta^{11}B$ value of -7.27‰ from the University of Southampton. This represents a $\delta^{11}B$ correction of 0.1 to 0.7 ‰.

line 237 – I think using Anand needs some real justification (see discussion/figure above) line 241 – an aside as you are not accounting for dissolution downcore, but it should really be parameterized as a function of Omega rather than depth. | line 246 – I think more details are need of the method you used for the pH corrected approach, and greater discussion of the results are warranted later. | Line 461 – why? We have now included a more complete approach by presenting various treatments of Mg/Ca and resulting $CO_2$ calculations. See detailed response above.

line 254 – how is the ALK range distributed around modern value. The uncertainty of 175 µmol/kg is distributed equally on either side of the central alkalinity value. This has been specified.

line 201 – with sacc or without sacc? | line 390 – sacc or no sacc? | line 524 – w/ or w/o sacc?
T. sacculifer was measured with sacc, but these leaching test results will be removed (see reply above and response to reviewer 1)
line 545 – really no TE data? Unfortunately not for T. sacculifer but available for G. ruber showing no changes in elements/Ca for G. ruber for different duration of leaching.

line 431 – I'm not sure this is the reason Pacific CaCO3 preservation increases during glacials, as there is a lot of evidence for reduced ventilation of the deep Pacific in glacials (e.g. Anderson et al 2019). An ALK increase due to the reduction of CaCO3 burial in the deep Atlantic seems more likely (e.g. Cartapanis et al 2018), but a lot of this discussion seems somewhat superfluous. It's enough to say that the fragmentation should give an indication of relative changes in dissolution and describe the patterns seen. We thank the reviewer for this added information and have rephrased the point about improved preservation during glacial. Even if this is not the main scope of the study we think it is important contextual information and have summarised the text as follow (line 521-528):

"The preservation cycle in the Pacific (Farrell and Prell, 1989), shows improved (poorer) preservation during glacial (interglacial) and this pattern seems to have developed after the mid Pleistocene transition (MPT) (Sexton and Barker, 2012). The origin of these cycles could be a combination of enhanced ventilation during glacials in the Pacific (Sexton and Barker, 2012), or increased burial due to enhanced global alkalinity following a decrease in burial in the Atlantic (Cartapanis et al., 2018). However glacial periods seem to have also been accompanied by a diminution in oxygenation in the deep Pacific (Anderson et al., 2019) that may have impacted preservation."

line 467 – in Gray et al 2018 we found no systematic Mg/Ca offset between morphotypes across the Indian and Atlantic. This reference has been added (line 582-584).

line 595 – worth noting optimized calibration is still very similar to other planktic calibrations i.e. it doesn't require something radically different to what we might expect. We agree with this statement and will specify this in the text. This highlights that a small change in the slope is enough to improves the fit to the ice core and diminishes the apparent correlation between high fragmentation and CO2 offset.

Text added line 737-741:

This analysis shows that a small change in the borate *G. ruber* $\delta^{11}$B calibration is enough to improve the fit to the ice core and diminishes the apparent correlation between high fragmentation and CO2 offset (Figure S14 ), and that uncertainty in the $\delta 11B_{foram-borate}$ calibration of Henehan et al. (2013) can – at least partly – explain the minor discrepancies we observe between $\delta^{11}$B-derived and ice core CO$_2$.

line 623 – the disequilibrium pump has garnered a lot of attention recently and is worth noting (e.g. Eggleston and Galbraith, 2018). Thank you for this input, this reference will be added (line 753).

figure 1 – better to use red-white-blue colour scheme for positive and negative anomalies. why not also add a map of modelled LGM-PI DpCO2 differences below with the core sites indicated? The colour scheme has been updated to colourblind-friendly in Figure S1. A reference to the delta pH map of Gray and Evans (2019) has been added. Figure s14 – why not plot on the other planktic lines to enable comparison with optimized line

This is a good point and the figure has been updated as follow (Figure R3, Figure S13 in the revised manuscript).

[Figure]

Figure R3. Comparison of the optimised $\delta^{11}$B foram/borate calibration with the published calibration slope of Henehan et al. (2013) along with other calibration of spinose foraminifera *T. sacculifer* (Martínez-Botí et al., 2015), *O. universa* (Henehan et al., 2016) and *G. bulloides* (Martínez-Botí et al., 2015)